# RETRIEVAL-AUGMENTED GENERATION WITH ESTIMATION OF SOURCE RELIABILITY

## ABSTRACT

Retrieval-augmented generation (RAG) addresses key limitations of large language models (LLMs), such as hallucinations and outdated knowledge, by incorporating external databases. These databases typically consult multiple sources to encompass up-to-date and various information. However, standard RAG methods often overlook the heterogeneous source reliability in the multi-source database and retrieve documents solely based on relevance, making them prone to propagating misinformation. To address this, we propose Reliability-Aware RAG (RA-RAG) which estimates the reliability of multiple sources and incorporates this information into both retrieval and aggregation processes. Specifically, it iteratively estimates source reliability and true answers for a set of queries without ground truth answers. Then, it selectively retrieves relevant documents from a few of reliable sources and aggregates them using weighted majority voting, where the selective retrieval ensures scalability while not compromising the performance. We also introduce a benchmark designed to reflect real-world scenarios with heterogeneous source reliability and demonstrate the effectiveness of RA-RAG compared to a set of baselines.

## 1 INTRODUCTION

Large language models (LLMs) have demonstrated remarkable capabilities across various tasks but are often limited by hallucinations and a lack of access to real-time information. Retrieval-augmented generation (RAG) offers a promising solution by integrating external databases, allowing LLMs to access more accurate and up-to-date information (Guu et al., 2020; Lewis et al., 2020). However, these databases often consist of content from multiple sources with varying levels of reliability, which can make RAG systems vulnerable to misinformation from unreliable sources. Standard RAG methods retrieve documents based on only relevance to the query, without considering the accuracy or trustworthiness of the information retrieved. As a result, RAG systems are prone to propagating misinformation. This issue is exacerbated by the advances of LLMs, which enable a massive production of plausible yet false documents and disables identifying the credibility of documents by linguistic features (Menczer et al., 2023; Augenstein et al., 2024; Hong et al., 2024; Zou et al., 2024; Shafran et al., 2024; Chaudhari et al., 2024; Chen & Shu, 2023).

Several recent efforts have sought to improve the robustness of RAG systems against misinformation (Pan et al., 2023; Weller et al., 2024; Xiang et al., 2024; Deng et al., 2024; Pan et al., 2024). However, these approaches have notable limitations. Deng et al. (2024) use LLMs to evaluate document reliability based on their internal knowledge, which is ineffective when LLMs indeed need to consult with external knowledge. Pan et al. (2023); Weller et al. (2024); Xiang et al. (2024) use counting-based methods such as majority voting (MV) or selecting responses that exceed a certain threshold. However, these methods are effective only when the portion of misinformation in the retrieved documents is minor, overlooking the heterogeneity in source reliability. Pan et al. (2024) propose a heuristic estimation, which categorizes the source reliability into two groups (low, high) based on a common reputation of the source. To learn how to aggregate responses according to source reliability, they generate labeled dataset to fine-tune LLMs. However, the reputation of sources, particularly social media and blogs, is often unclear and susceptible to manipulation. Additionally, a simple two-level credibility score cannot capture subtle differences in reliability and requires additional training to utilize this information.

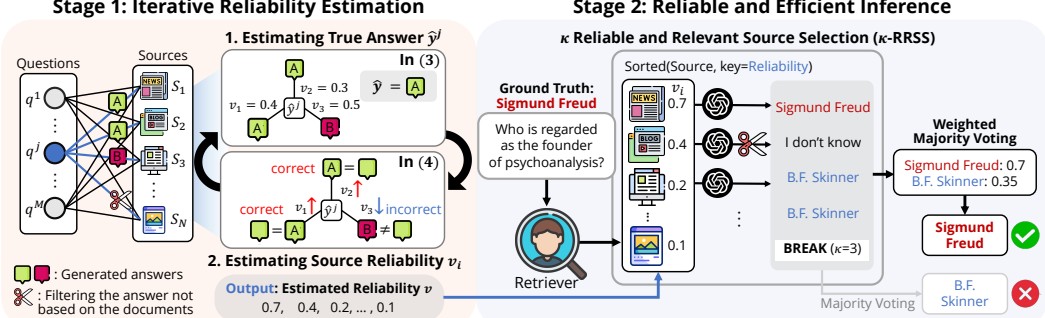

Figure 1: **Overview of the RA-RAG framework.** In the first stage, RA-RAG iteratively estimates the reliability of each source $v_i$ for $i \in [N]$ based on estimated true answers $\hat{y}^j$ for each question $j$, as outlined in equation (3) and (4). Based on the estimated reliability, in the second stage, the retriever selects $\kappa$ sources through the Reliable and Relevant Source Selection ($\kappa$-RRSS) process, detailed in Section 3.4. The final answer is determined using a weighted majority voting process, with the weights derived from the estimated reliability.

To overcome these limitations, we formalize a multi-source RAG framework based on weighted majority voting (WMV) that distinguishes between different sources in the database and integrates information from multiple sources given their reliability. However, as it is hard to know the source reliability in advance, we introduce Reliability-Aware RAG (RA-RAG), an effective method that estimates the reliability of each source and uses this information to guide the retrieval and aggregation process. RA-RAG operates in two stages illustrated in Figure 1. The first stage focuses on estimating source reliabilities for a set of queries with no labelings. Specifically, we devise an iterative reliability estimation that alternates estimating the true answers and the reliability of each source. In the second stage, RA-RAG aims at reliable and efficient inference, which aggregates documents from sources given the estimated source reliability while ensuring computational scalability even with numerous sources without compromising performance. To do so, we propose WMV with $\kappa$-reliable and relevant source selection ($\kappa$-RRSS), in which RA-RAG consults with only a small number of selective sources that are reliable with relevant documents.

RA-RAG also addresses two specific issues inherent in RAG systems: misalignment of responses, where the model generates answers based on internal knowledge rather than retrieved documents, and response variation, where semantically identical answers are phrased differently. These issues can hinder both the aggregation of responses and the estimation of source reliability. To overcome these challenges, RA-RAG incorporates a misalignment filtering mechanism to detect and exclude hallucinated responses, and a keyword-based system prompt to regularize response variation.

To evaluate the effectiveness of RA-RAG, we develop a realistic benchmark of multi-source RAG with heterogeneous source reliability, reflecting the complexity of real-world scenarios. Our benchmark contrasts with previous works (Weller et al., 2024; Xiang et al., 2024; Deng et al., 2024; Pan et al., 2024) that relied on setups with artificially injected misinformation into retrieved documents, as we construct an environment where the database consists of multiple sources with heterogeneous reliability. Our experimental results show that RA-RAG consistently outperforms various baselines, highlighting its robustness and effectiveness in accurately aggregating information from multiple sources.

We defer an extensive discussion of related works to Appendix A, while we summarise our main contributions as follows:

- We formalize the multi-source RAG (Section 2) and propose an effective approach, called RA-RAG (Section 3). It estimates the source reliability without additional labelings via the iterative reliability estimation (Section 3.3). Then, based on the estimated reliability, it efficiently retrieves reliable and relevant documents by $\kappa$-RRSS and robustly aggregates them with WMV (Section 3.4).

- We also address the inherent issues of RAG systems: variation and misalignment in the generated responses. To mitigate the issues, we devise the keyword-based system prompt (Section 3.1) and the misalignment filtration (Section 3.2).

- We construct the realistic benchmark of the multi-source RAG frameworks with diverse source reliability, which allows us to evaluate and analyze multi-source RAG systems (Section 4). Our code and datasets will be released.
- Our experimental results demonstrate that RA-RAG consistently outperforms a number of baselines, showcasing its robustness and effectiveness in accurately aggregating information from multiple sources, even in conflicting and unreliable information (Section 5).

## 2 PROBLEM FORMULATION

In this section, we formally describe the problem of retrieval-augmented generation (RAG) with a multi-source database. We introduce the standard RAG system in Section 2.1 and outline the multi-source RAG framework in Section 2.2, followed by a discussion of key challenges in Section 2.3.

### 2.1 RETRIEVAL-AUGMENTED GENERATION (RAG)

A typical RAG system generates a response $\hat{y}$ by retrieving relevant information from a database $\mathcal{D}$ using a retrieval mechanism $\mathcal{R}$, followed by a language model $\mathcal{G}$ to generate the final response. To be specific, consider a closed-ended query $q$ associated with a true answer $y$. Retriever $\mathcal{R}$ selects the top-$K$ most relevant documents from database $\mathcal{D}$ based on a similarity measure between query $q$ and each document $t \in \mathcal{D}$. The set of retrieved documents is denoted by $\mathcal{R}(q, \mathcal{D})$. Using the retrieval result $\mathcal{R}(q, \mathcal{D})$ and a system prompt $\mathcal{P}$, language model $\mathcal{G}$ generates a response $\hat{y}$, which can be represented as follows: $\hat{y} = \mathcal{G}(q, \mathcal{R}(q, \mathcal{D}), \mathcal{P})$ . In this framework, system prompt $\mathcal{P}$ guides language model $\mathcal{G}$ to generate response $\hat{y}$ based on the retrieval result $\mathcal{R}(q, \mathcal{D})$. A critical limitation arises when the retrieved documents come from unreliable sources, as the similarity measure used in retrieval does not account for the truthfulness or reliability of the information. This becomes particularly problematic when database $\mathcal{D}$ includes documents from multiple sources with different levels of reliability, such as news articles, Wikipedia, or social media. Such a multi-source database is often necessary to cover up-to-date and diverse content. Although prompts can attempt to filter out unreliable information based on linguistic features, this is not always reliable. With the growing sophistication of language models, generating plausible yet false information—such as fake news—has become easier (Menczer et al., 2023; Augenstein et al., 2024). This underscores the need for additional mechanisms to assess the reliability of retrieved documents, beyond their mere relevance to the query.

### 2.2 MULTI-SOURCE RAG

To address the vulnerability to misinformation in retrieval, we propose a multi-source RAG framework that explicitly distinguishes between the sources of documents. Let $N$ be the number of distinct sources contributing to database $\mathcal{D}$. We partition the database as $\mathcal{D} = \bigcup_{i=1}^{N} \mathcal{S}_i$, where $\mathcal{S}_i$ is the set of documents originating from source $i \in [N]$. Such a partition of dataset $\mathcal{D}$ enables the system to account for the reliability of each document's source, based on weighted majority voting (WMV). For a given query $q$, let $\tilde{y}_i = \mathcal{G}(q, \mathcal{R}(q, \mathcal{S}_i), \mathcal{P})$ represent the response generated using documents exclusively from source $i$. Once the probability that a retrieved document from source $i$ is correct is estimated as $v_i$, and a set of candidate responses $\mathcal{M}$ is obtained from $\tilde{y}_i$'s, we can apply WMV to aggregate the responses:

$$\hat{y} = \arg\max_{u \in \mathcal{M}} \sum_{i \in [N]} v_i \mathbb{1}(\tilde{y}_i = u) \ . \tag{1}$$

If all sources are assumed to have equal reliability, this reduces to majority voting (MV) that selects the most consensus among $\tilde{y}_i$'s. However, WMV is superior to MV when the reliability of each source $v_i$ can be properly estimated. Hence, the key components in a multi-source RAG framework are (i) the reliability estimation for $v_i$'s and (ii) the response aggregation of $\tilde{y}_i$'s based on WMV.

### 2.3 CHALLENGES IN MULTI-SOURCE RAG

To develop an effective mechanism for reliability estimation and response aggregation in the multi-source RAG framework, we need to address four key challenges: (i) response variations, (ii) misaligned response, (iii) limited access to ground truth, and (iv) scalability in the number of sources.

**Response variations.** Even when focusing on closed-ended queries, language models often generate semantically identical answers with paraphrasing, complicating the application of weighted majority voting (WMV). For example, in response to the question, "What is the largest planet in our solar system?", the model might generate "The largest planet in our solar system is Jupiter" or "Jupiter is the largest planet in our solar system." These responses convey the same meaning, but if treated as different outputs, they can reduce the effectiveness of WMV. Therefore, techniques for normalizing or aligning semantically equivalent responses are essential to ensure that the aggregation process accurately reflects consensus.

**Misaligned responses.** Since the reliability of sources is inferred based on the outputs generated by the language model, the model must generate responses strictly from the documents retrieved from each source. However, language models often produce hallucinations or responses influenced by their internal knowledge, particularly when dealing with complex queries or ambiguous contextual information (Kaddour et al., 2023; Ji et al., 2023; Xie et al., 2024; Kortukov et al., 2024; Xu et al., 2024). These misaligned responses complicate the reliability estimation process by distorting the relationship between the response and the quality of the retrieved documents. Thus, it is necessary to develop a filtering mechanism to exclude responses that do not rely on the retrieved documents, ensuring that the answers are grounded in the provided information.

**Limited access to ground truth.** A straightforward reliability estimation can be conducted with the true labels for a number of queries. However, acquiring the labels is costly and time-intensive, particularly when dealing with unverified, up-to-date information. Hence, we consider a set of queries with no labels and devise an iterative algorithm that alternates between estimating both the reliability of sources and the true answers (Section 3.3).

**Scalability in the number of sources.** In a multi-source RAG framework, retrieving documents from a large number of sources can lead to significant computational overhead. Generating responses $\tilde{y}_i$ for every source may become impractical as the number of sources increases. To ensure scalability in real-world applications, it is critical to use strategies that reduce computational costs, such as optimizing retrieval processes or limiting the number of sources considered during inference. Effective solutions will enable multi-source RAG systems to scale efficiently without compromising performance (Section 3.4).

## 3 RELIABILITY-AWARE RETRIEVAL-AUGMENTED GENERATION (RA-RAG)

To mitigate the impact of misinformation on RAG systems, we propose the Reliability-Aware Retrieval-Augmented Generation (RA-RAG), which addresses the challenges of the multi-source RAG framework outlined in Section 2.3. RA-RAG operates in two stages: (i) iterative reliability estimation (Section 3.3) and (ii) reliable and efficient inference (Section 3.4). In the first stage, the iterative algorithm given a set of queries with no labels estimates the reliability of sources before deploying the RAG system. In the second stage, a WMV mechanism combined with $\kappa$-reliable and relevant source selection ($\kappa$-RRSS) uses the estimated reliability to retrieve documents and effectively aggregate the information. In advance of describing the two stages of RA-RAG, we first present two common components used for both stages: keyword-based system prompt (Section 3.1) and misalignment filtration (Section 3.2). They are mainly to address the problems of variation and misalignment in the language model's responses. The detailed algorithm of RA-RAG can be found in Appendix B, and the code will be distributed.

### 3.1 KEYWORD-BASED SYSTEM PROMPT

Even for closed-ended queries that elicit simple answers, such as "What is the largest planet in our solar system?", the language model often generates responses of the same semantic meaning but in various forms. This decreases the effectiveness of response aggregation with WMV and thus disturbs the reliability estimation as well. To address this, we employ a system prompt to ensure that the language model generates keyword-based answers without unnecessary details. For example, the keyword-based response to the aforementioned query would be "Jupiter", rather than a more detailed response, "Jupiter is the largest planet in our solar system.". In addition, the prompt also guides the language model to say IDK (which stands for "I don't know" and indicates the abstention) when irrelevant documents are retrieved. The detailed prompt is provided in Appendix G. This

keyword-based answer allows us to effectively aggregate the LLMs response for WMV while reducing hallucinations to generate responses irrelevant to retrieval results. We note that Xiang et al. (2024) use a post-processing method with the spaCy library (Honnibal et al., 2020) to extract keywords like adjectives, adverbs, and other elements from LLM outputs. However, we observe that prompt engineering alone can effectively generate the desired keyword-based outputs without the need for additional processing. The detailed experimental results are provided in Appendix E.

## 3.2 MISALIGNMENT FILTRATION

Although the system prompt directs the language model to generate $\tilde{y}_i$ based on the retrieval result $\mathcal{R}(q, \mathcal{S}_i)$, responses irrelevant to the retrieval result are often generated. This can distract the reliability estimation and WMV, as discussed in Section 2.3. To mitigate this issue, we further employ a filtration function $f_{\text{align}}$ to detect misaligned responses and replace them with IDK, indicating the abstention. In this paper, we use the ROUGE-1 precision score (Lin, 2004) which computes the ratio of the number of unigrams in response $\tilde{y}_i$ that appear also in retrieval result $\mathcal{R}(q, \mathcal{S}_i)$. A low score implies that the response is irrelevant to the retrieved documents. Specifically, $f_{\text{align}}$ replaces $\tilde{y}_i$ with IDK if the score for a response is below 0.9 and keep $\tilde{y}_i$ otherwise. Given that the responses are in a keyword-based form, the ROUGE-1 precision score effectively captures how well the words in the response match the provided context. As a result, we can obtain a refined set of answer candidates:

$$\mathcal{M}_{\text{filtered}} = \{f_{\text{align}}(\tilde{y}_i, \mathcal{R}(q, \mathcal{S}_i)) \mid i \in [N]\} . \tag{2}$$

We demonstrate the effectiveness of $f_{\text{align}}$ based on ROUGE-1 precision in Section 5.3. However, our choice of $f_{\text{align}}$ based on the ROUGE-1 score can be replaced with asking the relevance of the response to the retrieved result to the language model, while it requires substantial cost as well as the risk of hallucination.

## 3.3 STAGE 1: ITERATIVE RELIABILITY ESTIMATION

To estimate the reliability of sources and effectively aggregate the outputs, we utilize the WMV method proposed by Li & Yu (2014), which is a simple and effective approach for aggregating crowdsourced labels in classification tasks, applied to the filtered outputs. Given the set $\{q^j \mid j \in [M]\}$ of $M$ queries, the iterative reliability estimation is described as follows:

- Step 0. We initialize the source weights $v_i = 1$ for all sources $i \in [N]$ and repeat Step 1 to Step 2 until $v_i$'s converge.

- Step 1. We estimate each answer $\hat{y}^j$ for each $j \in [M]$ using WMV:

$$\hat{y}^j = \arg\max_{u \in \mathcal{M}^j_{\text{filtered}}} \sum_{i \in [N]} v_i \mathbb{1}(\tilde{y}^j_i = u) , \tag{3}$$

where $\tilde{y}^j_i = \mathcal{G}(q^j, \mathcal{R}(q^j, \mathcal{S}_i), \mathcal{P}_{\text{keyword}})$ is a response to $q^j$ based on documents from source $i$ and $\mathcal{M}^j_{\text{filtered}} = \{f_{\text{align}}(\tilde{y}^j_i, \mathcal{R}(q, \mathcal{S}_i)) \mid i \in [N]\}$ is the filtered candidates of responses to $q^j$.

- Step 2. Given the estimated $\hat{y}^j$'s, we estimate source reliability $\hat{w}_i$ for each $i \in [N]$ as follows:

$$\hat{w}_i = \frac{\sum_{j=1}^{M} \mathbb{1}\left(f_{\text{align}}(\tilde{y}^j_i, \mathcal{R}(q^j, \mathcal{S}_i)) = \hat{y}^j\right)}{\sum_{j=1}^{M} \mathbb{1}\left(f_{\text{align}}(\tilde{y}^j_i, \mathcal{R}(q^j, \mathcal{S}_i)) \neq \text{IDK}\right)} . \tag{4}$$

Then, we normalize $\hat{w}_i$ as $v_i = N\hat{w}_i - 1$ giving larger weights to reliable sources and smaller weights to unreliable sources. This leads to more accurate estimates of $w_i$. Following the approach of Li & Yu (2014), which uses the number of labels $L$ as a scaling factor, we use $N$, the number of sources, because the maximum possible number of responses occurs when each source provides a distinct answer. However, in real-world applications, it is unlikely that all sources will offer completely distinct responses, especially when $N$ is large. Thus, $N$ can be reasonably limited to a manageable size.

- Step 3. With the converged weights $v_i$, we compute the refined estimated answers using the WMV, i.e., $\hat{y}^j = \arg\max_{u \in \mathcal{M}^j_{\text{filtered}}} \sum_{i \in [N]} v_i \mathbb{1}(\tilde{y}_i = u)$ for all $j \in [M]$.

### 3.4 STAGE 2: RELIABLE AND EFFICIENT INFERENCE

In real-world applications, integrating information from all sources can cause a significant computational burden, especially when the number of sources is large. The WMV process, $\hat{y}^j = \arg\max_{u \in \mathcal{M}^j_{\text{filtered}}} \sum_{i \in [N]} v_i \mathbb{1}(\tilde{y}^j_i = u)$, becomes increasingly expensive as $N$ grows. For better scalability, we select a subset of sources, denoted as $\mathcal{W}$, where $|\mathcal{W}| < N$. Then, the WMV process is applied using $\mathcal{W}$: $\hat{y}^j = \arg\max_{u \in \mathcal{M}^j_{\text{filtered-}\mathcal{W}}} \sum_{i' \in \mathcal{W}} v_{i'} \mathbb{1}(\tilde{y}_{i'} = u)$, where $\mathcal{M}^j_{\text{filtered-}\mathcal{W}} = \{f_{\text{align}}(\tilde{y}^j_{i'}, \mathcal{R}(q, \mathcal{S}_{i'})) \mid i' \in \mathcal{W}\}$. A simple way to select $\mathcal{W}$ is to choose the $\kappa$ most reliable sources. However, high reliability alone doesn't always mean these sources will contain relevant documents for the given query, which can lead to performance degradation. Therefore, it's important to consider both reliability and relevance.

To achieve this, we identify the $\kappa$ most reliable and relevant sources. We start by ordering the sources by reliability and then check if they contain documents relevant to the query. This process continues until we find $\kappa$ sources, which we call the $\kappa$-Reliable and Relevant Source Selection ($\kappa$-RRSS). Although similarity scores from the retrieval process can suggest relevance, they are often noisy, as high scores may just reflect shared words without capturing the actual context. Instead, we use the LLM's responses and apply a filtering method, $f_{\text{align}}$, to assess relevance. If the filtered LLM response is "I don't know", we consider the source as lacking relevant information. Once the $\kappa$ sources are selected, we use their outputs in the WMV process to generate the final answers.

## 4 BENCHMARK OF MULTI-SOURCE RAG

We construct a benchmark designed to assess performance in environments where the database contains sources with heterogeneous reliability, which reflect the complexities of real-world scenarios. Each source $S_i$ is characterized by two parameters: (1) the probability $r_i$ of containing the relevant documents to a given question (related to the corpus size), and (2) its reliability $p_i$, representing the probability that $S_i$ contains factual documents. We use two types of priors to model the reliability $p_i$: the *beta* prior and the *spammer-hammer* prior. For the *beta* priors, $p_i$ is sampled from Beta $\left(2\bar{w}/1-\bar{w}, 2\right)$, which has an expected mean of $\bar{w}$. This enables the simulation of scenarios with sources that have heterogeneous reliability. For the *spammer-hammer* prior, $p_i$ is set to either $0.1$ or $0.9$, where $p_i = 0.1$ represents a *spammer*, an unreliable source that provides mostly incorrect information for the given query and $p_i = 0.9$ represents a *hammer*, a reliable source that provides mostly accurate information.

To construct the corpus for each source $S_i$ based on this framework, we use three datasets: Natural Questions (NQ) (Kwiatkowski et al., 2019), HotpotQA (Yang et al., 2018), and TriviaQA (TQA) (Joshi et al., 2017). For the HotpotQA dataset, we use only single-hop queries. We focus on closed-ended queries because open-ended queries (e.g., "Describe the various uses of forests to human beings" from the NQ dataset) often lack definitive answers, making aggregation from multiple sources difficult. To identify closed-ended queries, we first use GPT-4o-mini (OpenAI, 2024) to filter out open-ended queries and then manually review any remaining ones. The filtering prompt is provided in Appendix F.1. Due to computational constraints, we select 1,600 queries per dataset: 200 for reliability estimation and 1,400 as test data. The corpus for each source $S_i$ is generated through the following steps:

1. **Collecting factual documents**: We first collect documents containing the correct answers from the Wikipedia corpus using Contriever (Izacard et al., 2022) for the NQ and TQA datasets. For the HotpotQA dataset, we use the contexts provided within the dataset itself.

2. **Generating diverse factual information**: To generate diverse factual information that conveys the same meaning but in different expressions, we use GPT-4o-mini to paraphrase the collected documents, creating 9 documents for each query. This diversity makes it more challenging to aggregate the LLM's outputs.

3. **Generating diverse misinformation**: Unlike classification tasks with predefined label sets, incorrect answers can vary infinitely in query-answering tasks. To simplify our experiment, we use GPT-4o-mini to generate 9 distinct incorrect answers for each query and then create three corresponding documents for each incorrect answer using GPT-4o-mini.

4. **Constructing the corpus for** $S_i$: The corpus for each source $S_i$ is generated independently, based on its $r_i$ and $p_i$. If $S_i$ contains relevant documents for a given query (as determined by $r_i$), the truthfulness of these documents is decided by $p_i$. If $S_i$ is assigned to provide factual information, it randomly selects three documents from pool of the previously generated factual documents. Conversely, if $S_i$ is designated to provide misinformation, it randomly choose one of the nine incorrect answers and includes the corresponding three misinformation documents generated earlier. Since the corpus for each source is constructed independently, each source contains a different set of knowledge. For instance, for a given query, $S_i$ may have relevant documents, while $S_j$ may not, where $i \neq j$ and $i, j \in [N]$.

The specific prompts used to generate the data are detailed in Appendix F.

## 5 EXPERIMENTS

In this section, we conduct comprehensive experiments on the benchmarks to evaluate the effectiveness of RA-RAG. Detailed descriptions of the experimental setup can be found in Section 5.1, and the results are presented in Section 5.2. Additionally, we perform ablation studies on the individual modules within RA-RAG to evaluate its overall effectiveness, as discussed in Section 5.3.

### 5.1 EXPERIMENTAL SETTINGS

**Models.** For the LLMs, we use Llama3-8B Instruct (Dubey et al., 2024), Phi3-mini Instruct (Abdin et al., 2024), and GPT-4o mini (OpenAI, 2024). As the retriever, we utilize Contriever (Izacard et al., 2022), which calculates similarity scores by taking the dot product between the embedding vectors of a query and documents in the database.

**Baselines.** We evaluate our framework against the following baselines:

- **Oracle WMV**: Oracle Weighted Majority Voting aggregates outputs using the true reliability values of each source, representing an ideal scenario where the source reliability is available.
- **WMV**: Weighted Majority Voting aggregates outputs from all sources using the estimated reliability values, i.e., Weighted Majority Voting excludes the RRSS process from RA-RAG.
- **MV**: Majority Voting aggregates outputs by assigning equal weight to each response.
- **Naive RAG**: Naive RAG follows the traditional RAG approach, retrieving documents and generating outputs without considering the reliability of the sources in the database.
- **Naive LLM**: Naive LLMs generate outputs for the given query without using a retriever.

**Settings for multi-source RAG.** In our multi-source RAG setup, we retrieve the top 3 documents from each source and select 4 sources for the $\kappa$-RRSS process. For Naive RAG, we retrieve the top-10 documents, as multi-source RAG typically handles a larger volume of information. When constructing sources for the multi-source RAG benchmark, we set the beta prior with $\bar{w} = 0.6$. To simplify the setup, we assign $r_i = 0.6$ for all sources. Due to computational constraints, the number of sources is limited to between 3 and 9. We use Exact Match (EM) (Rajpurkar et al., 2016) as the evaluation metric, with all experimental results averaged over 10 random trials.

### 5.2 MAIN RESULTS

**Beta prior.** We conduct experiments using a *beta* prior across different numbers of sources to demonstrate the effectiveness of our method with heterogeneous reliability. Figure 2 shows that our RA-RAG framework demonstrates significant performance improvements, outperforming both MV and Naive RAG. Notably, our RA-RAG, which uses $\kappa = 4$ for $\kappa$-RRSS (the subset of sources), achieves performance comparable to WMV, which aggregates information from all available sources. The performance of WMV closely matches the oracle WMV, demonstrating that our approach effectively estimates source reliability. In contrast, the Naive RAG exhibits poor performance when source reliability varies, leading to a high likelihood of retrieving a mixture of factual information and misinformation, for both the NQ and TQA datasets. This inconsistency results in conflicting knowledge, complicating LLMs' ability to generate accurate outputs.

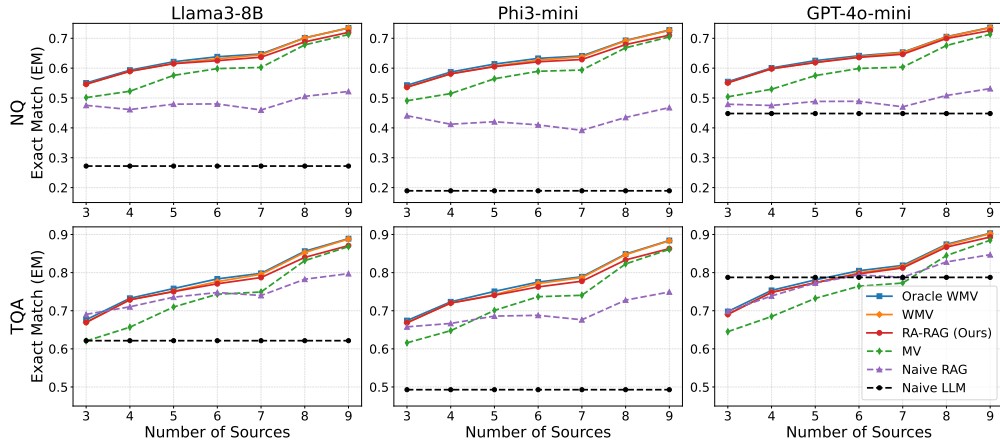

Figure 2: Exact Match against the number of sources, comparing the heterogeneous reliability via *beta* priors on the NQ and TQA datasets across the Llama3-8B, Phi3-mini, and GPT-4o-mini language models. See Figure 8 in Appendix C.1 for the HotpotQA dataset.

**Question**: The gulf stream the world's fastest ocean current flows along the western side of this water body
**Ground Truth (GT)**: atlantic ocean

| Multi-Source Outputs | southern ocean | atlantic ocean | I don't know | indian ocean | southern ocean | gulf of mexico |
|---|---|---|---|---|---|---|
| **True Reliability** | 0.39 | **0.97** | 0.55 | 0.78 | 0.53 | 0.79 |
| **Estimated Reliability** | 0.36 | **0.92** | 0.56 | 0.82 | 0.54 | 0.76 |

**MV Answer**: southern ocean     **RA-RAG Answer**: atlantic ocean

Figure 3: A qualitative comparison between MV and RA-RAG on the NQ dataset. Additional examples are available in Appendix D.

We provide a qualitative example in Figure 3 to illustrate how RA-RAG effectively estimates source reliabilities and improves answer accuracy. While MV selects "southern ocean" based on response frequency, RA-RAG correctly identifies "atlantic ocean" by leveraging well-estimated reliabilities that align closely with true values. This example highlights RA-RAG's robustness in overcoming conflicting or inaccurate information by focusing on more reliable sources, ensuring more accurate predictions.

**Spammer-hammer prior.** To evaluate the robustness of our method in the presence of spammers in the database, we conduct experiments using the *spammer-hammer prior* with a total of 9 sources, using Llama3-8B on the NQ dataset. The experimental results for TQA and HotpotQA datasets are provided in Appendix C.2. As shown in Figure 4, our RA-RAG framework demonstrates robustness against spammers, while the performance of Naive RAG degrades significantly as the number of spammers increases. When the number of spammers exceeds five, MV performs worse than Naive RAG. This decline results from the dominance of misinformation from spammers, which leads MV to select incorrect answers.

### 5.3 ABLATION STUDIES AND ANALYSIS

**Effectiveness of filtering with $f_{\text{align}}$.** We conduct an ablation study to evaluate the necessity of $f_{\text{align}}$ for accurate reliability estimation, across different types of retrieved documents: factual, misinformation, and irrelevant. Table 1 shows the proportions of answers both without (w/o) and with (w/) filtering, categorized by answer types: correct, incorrect, IDK (I don't know), and hallucination, based on 1,600 queries in the TQA dataset, using a single source with $p_i = 0.5$ and $r_i = 0.5$. Further results for other datasets and models are in Appendix C.5. Table 1 shows that LLM misalignment occurs when the documents contain misinformation or are irrelevant, resulting in correct or hallucinated answers that are not based on the retrieved documents. In particular, in cases of irrelevant documents, LLMs often generate either correct answers or hallucinations. However, after applying the $f_{\text{align}}$, these misaligned responses (marked in blue) are significantly reduced. Additionally, the

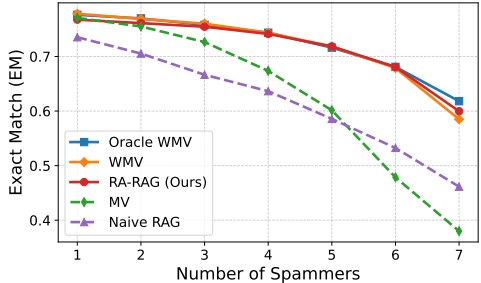

Figure 4: EM against the number of spammers, comparing robustness using *spammer-hammer* priors on the NQ dataset.

Figure 5: EM for different numbers of $\kappa$ on the NQ dataset.

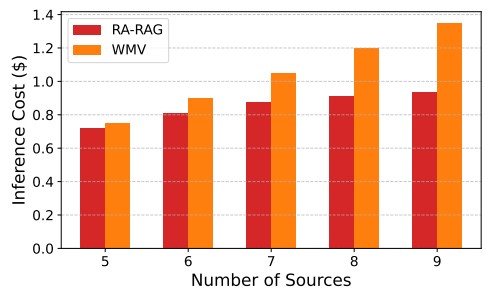

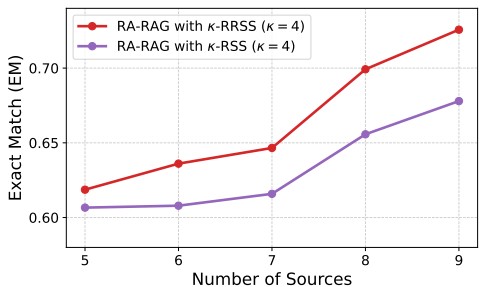

Figure 6: Comparison of inference costs between RA-RAG and WMV using the GPT-4o-mini.

Figure 7: Comparison of $\kappa$-RRSS and $\kappa$-RSS ($\kappa = 4$) across different number of sources.

Table 1: Percentage of type of answers by retrieved document type in the filtering with $f_{\text{align}}$ ablation study on Llama3-8B and TQA dataset.

| Types of Answers | Filtering ($f_{\text{align}}$) | Types of Retrieved Documents | | |
|---|---|---|---|---|
| | | **Factual** | **Misinformation** | **Irrelevant** |
| **Correct** | w/o | 92.82% | 5.43% | 25.55% |
| | w/ | 92.82% | 0.82%(-4.61) | 4.30%(-21.25) |
| **Incorrect** | w/o | - | 81.52% | - |
| | w/ | - | 81.52% | - |
| **IDK** | w/o | 0.00% | 4.89% | 55.52% |
| | w/ | 0.00% | 11.68%(+6.79) | 86.36%(+30.84) |
| **Hallucination** | w/o | 7.18% | 8.15% | 18.92% |
| | w/ | 7.18% | 5.98%(-2.17) | 9.34%(-9.58) |

Table 2: Ablation study on the risk of distorted reliability estimation without $f_{\text{align}}$.

| Method | EM |
|---|---|
| Oracle WMV | 0.549 |
| Ours (w/) | 0.543 |
| Ours (w/o) | 0.465 |
| MV | 0.449 |

increase in IDK responses (marked in red) indicates that filtering effectively mitigates the influence of irrelevant documents and misinformation.

**Risk of distortion of reliability estimation without filtering.** As highlighted in our analysis, LLMs frequently generate either correct answers or hallucinations when processing irrelevant documents. This issue becomes particularly problematic when dealing with low-reliability sources that have a small corpus. As the corpus size shrinks, i.e., contains fewer relevant documents, more queries are needed to accurately estimate the reliability of these sources. However, this can lead to LLMs producing a large number of correct answers or hallucinations, potentially resulting in an overestimation of the sources' reliability. To illustrate this risk, we conduct experiments using the *spammer-hammer prior*. Specifically, we experiment with five sources from the TQA dataset using Llama3-8B: four spammers with $r_i = 0.1$ and one hammer with $r_i = 0.6$, using 800 queries for reliability estimation. As illustrated in Table 2, without filtering, the estimated weights are distorted, giving more weight to the spammers and leading to poor performance. However, applying filtering effectively mitigates this distortion in reliability estimation, yielding performance close to the Oracle WMV.

**Effectiveness of $\kappa$-RRSS.** To determine the optimal value of $\kappa$ for $\kappa$-RRSS, we conduct ablation studies using different values of $\kappa$ with 9 sources on the NQ dataset, employing Llama3-8B. As

shown in Figure 5, when $\kappa = 4$, RA-RAG outperforms MV and shows a slight performance drop compared to WMV, which uses all available sources. This result, with $\kappa$ being less than half the total number of sources, shows that selecting a small subset of sources can achieve performance close to using all sources. The results for additional datasets are provided in Appendix C.3. In addition, to quantitatively evaluate the efficiency of $\kappa$-RRSS, we assess the inference cost using GPT-4o mini by the number of sources. As shown in Figure 6, RA-RAG incurs lower inference costs than WMV, which aggregates all sources, and the efficiency of our RA-RAG framework becomes increasingly evident as the number of sources grows.

To analyze the importance of considering relevance in $\kappa$-RRSS, we evaluate a variant: $\kappa$-RSS (Reliable Source Selection), which selects $\kappa$ sources based solely on reliability scores in descending order, without checking whether the sources contain relevant documents on NQ dataset. As shown in Figure 7, using $\kappa$-RSS leads to significant performance degradation, since high reliability alone does not guarantee that the selected sources will contain documents relevant to the given query.

## 6    CONCLUSION

In this paper, we introduce RA-RAG, an effective multi-source RAG mechanism that estimates source reliability without using true labels and efficiently aggregates information through WMV, addressing the inherent problems of RAG systems. We also present a comprehensive benchmark that reflects real-world scenarios with diverse source reliability. Our experimental results show that RA-RAG consistently outperforms a set of baselines, demonstrating its robustness and effectiveness even when handling conflicting and unreliable information.

**Limitations and future directions.** Despite RA-RAG's robustness, several challenges remain: (1) While aggregating keyword-based LLM responses using a system prompt is effective and does not require additional post-processing or modules, this approach has limitations, such as difficulty in handling homonyms and varied expressions, which make consistent aggregation challenging. Therefore, more advanced approaches, such as using LLMs capable of capturing semantic meaning, are needed for more general and reliable aggregation. We discuss advanced aggregation methods we have explored in Appendix I. (2) In this work, we primarily focus on short-form generation tasks. However, RA-RAG can be extended to long-form generation tasks by using text decomposition, where long-form responses are broken down into a series of short-form responses to evaluate long-form factuality (Min et al., 2023; Wei et al., 2024; Farquhar et al., 2024). This decomposition reduces the long-form generation task into multiple short-form tasks, allowing our approach to be applied. Further exploration and details of this extension is provided in Appendix J. (3) While $f_{align}$ effectively filters misaligned responses, it struggles to detect incorrect answers based on retrieved documents. More reliable evaluation methods, such as applying modules from advanced RAG, are necessary. Since our approach does not require training, it can be applied in a plug-and-play manner alongside advanced RAG systems. (4) While we assume the availability of queries to estimate source reliability, real-world applications often require generating these queries, which adds complexity to accurately estimating source reliability.

## 7    REPRODUCIBILITY STATEMENT

As mentioned, we will release the source code and datasets for the multi-source RAG benchmark and the set of method including RA-RAG and the other baselines. Our experiments have been produced using LLama3-8B Instruct, Phi3-mini Instruct and ChatGPT-4o-mini from July to September 2024. We will also distribute the intermediate outputs of the LLMs so that every evaluation in this work can be reproducible.

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

# A    RELATED WORKS

**Vulnerabilities of RAG to misinformation.** Retrieval-Augmented Generation (RAG) systems address inherent limitations of LLMs, such as hallucinations and a lack of access to up-to-date knowledge (Lewis et al., 2020; Guu et al., 2020), by integrating external databases through retrievers with LLMs. However, standard RAG systems often fall short in handling more complex real-world scenarios. To address these challenges, several advanced RAG systems (Jiang et al., 2023; Asai et al., 2024) have been developed, incorporating multiple modules to enhance their performance. Despite these advancements, concerns about misinformation threats within RAG frameworks are growing, as misinformation in retrieved results can lead LLMs to generate unreliable outputs (Pan et al., 2023; Chen et al., 2024; Greshake et al., 2023; Hong et al., 2024). Furthermore, recent studies (Zhong et al., 2023; Zou et al., 2024; Shafran et al., 2024; Chaudhari et al., 2024) have highlighted the vulnerability of RAG systems to data poisoning attacks, where adversaries craft a small number of malicious documents to disrupt the system's reliability. These attacks exploit a key weakness in RAG systems that rely on relevance scores for retrieval; by creating malicious documents with high similarity scores to target questions, adversaries can deceive the RAG system into retrieving misleading information. Notably, Zou et al. (2024) demonstrated that even advanced RAG systems are highly susceptible to such data poisoning attacks.

**Robust RAG against misinformation threat.** In response to misinformation threats in RAG systems, several robust methods have been proposed. Pan et al. (2023) uses majority voting to enhance output reliability, while Weller et al. (2024) employs query augmentation to retrieve diverse documents and evaluates the frequency of generated outputs against the retrieved content. Xiang et al. (2024) adopts an isolate-then-aggregate strategy, generating LLM responses for each passage separately and then aggregating them securely to produce robust outputs. Deng et al. (2024) assigns heuristic reliability scores to the document based on source reputation and applies prompt engineering to prioritize documents from reputable sources, whereas Pan et al. (2024) uses GPT's internal knowledge to generate reliability scores of the document and adjust attention weights accordingly. However, these methods heavily depend on heuristics, with Weller et al. (2024); Pan et al. (2023); Xiang et al. (2024) being effective only when the true documents are the majority of the retrieved results. Moreover, heuristic approaches for determining the reliability scores of documents are often impractical in real-world scenarios, as the reliability of sources such as social media or blogs are often unknown or uncertain, and LLMs cannot rely on internal knowledge to assess up-to-date information. In contrast, our approach systematically estimates source reliability within the database and efficiently aggregates information from multiple sources, resulting in more robust and reliable outcomes.

**Robust answer aggregation.** In many AI systems, gathering information from diverse sources is essential for tasks such as data labeling, knowledge retrieval, and enhancing large language model (LLM) performance. A central challenge in these tasks is how to aggregate potentially conflicting information from various sources, especially when the quality and reliability of these sources are variable. The most straightforward strategy is majority voting (MV), which assigns the most frequent label or answer across sources. However, MV can be error-prone, especially when dealing with sources of varying reliability. To address this, studies such as Karger et al. (2011); Liu et al. (2012); Yue et al. (2014); Li & Yu (2014); Aydin et al. (2014); Li et al. (2016); Geng et al. (2020) have proposed weighted majority voting (WMV) approaches that account for source reliability, leading to more accurate aggregation of information.

Robust aggregation also plays a crucial role in enhancing the reliability of LLM outputs, particularly in recent complex reasoning tasks. For instance, Wang et al. (2023) introduces a method that samples diverse reasoning outputs in CoT (Wei et al., 2022) and aggregates the final output through majority voting. Similarly, Zhou et al. (2023); Wan et al. (2024) calculate confidence scores for each CoT output to perform weighted majority voting, thereby improving the robustness of answers. Further, Chen & Li (2024) adopts a reasoning roll-back strategy in Tree-of-Thoughts (ToT) (Yao et al., 2024) and applies weighted majority voting to produce the final output.

# B   DETAILED ALGORITHM FOR RA-RAG

---

**Algorithm 1** First Stage: Iterative Reliability Estimation

---

1: **Input:** multi-source database $\mathcal{D}_{\text{multi}} = \bigcup_{i=1}^{N} \mathcal{S}_i$, the number of queries $M_1$,
   filtering function $f_{\text{align}}$, IDK="I don't know"
2: **Output:** estimated weights $\{v_1, \dots, v_N\}$
3: $\mathcal{M}^j = \{f_{\text{align}}(\tilde{y}_i^j, \mathcal{R}(q^j, \mathcal{S}_i)) \mid i \in [N]\}$, where $\tilde{y}_i^j = \text{LLM}(q^j, \mathcal{R}(q^j, \mathcal{S}_i), \mathcal{P}_{\text{keyword}})$
4: **Initialization**: $v_i = 1, \forall i \in [N]$
5: **repeat**
6: $\quad \hat{y}^j = \arg\max_{u \in \mathcal{M}^j} \sum_{i \in [N]} v_i \mathbb{1}(f_{\text{align}}(\tilde{y}_i^j, \mathcal{R}(q^j, \mathcal{S}_i)) = u), \forall j \in [M_1]$
7: $\quad \hat{w}_i \leftarrow \frac{\sum_{j=1}^{M} \mathbb{1}(f_{\text{align}}(\tilde{y}_i^j, \mathcal{R}(q^j, \mathcal{S}_i)) = \hat{y}^j)}{\sum_{j=1}^{M} \mathbb{1}(f_{\text{align}}(\tilde{y}_i^j, \mathcal{R}(q^j, \mathcal{S}_i)) \neq \text{IDK})}, \forall i \in [N]$
8: $\quad v_i \leftarrow N\hat{w}_i - 1, \forall i \in [N]$
9: **until** convergence or reaches $T$ iterations.

---

**Algorithm 2** Second Stage: Reliable and Efficient Inference

---

1: **Input:** multi-source database $\mathcal{D}_{\text{multi}} = \bigcup_{i=1}^{N} \mathcal{S}_i$, the number of queries $M_2$, Retriever $\mathcal{R}$, esti-
   mated weights $\{v_1, \dots, v_N\}$, the number of reliable and relevant sources $K < N$
2: **Output:** $\{\hat{y}^j \mid j \in [M_2]\}$
3: Sort the sources $\{S_1, \dots, S_N\}$ based on their estimated weights $\{v_1', \dots, v_N'\}$ in descending
   order to get $\{\mathcal{S}_1', \dots, \mathcal{S}_N'\}$ such that $v_1' \geq v_2' \geq \dots \geq v_N'$
4: **for** $j = 1$ to $M_2$ **do**
5: $\quad$ Construct $\kappa$-RRSS $\leftarrow \{i' \mid f_{\text{align}}(\tilde{y}_{i'}^j, \mathcal{R}(q^j, \mathcal{S}_{i'}')) \neq \text{IDK}, \ i' \in [N]\}$ with up to $K$ elements,
   i.e., $|\kappa\text{-RRSS}| = k$
6: $\quad$ Construct $\mathcal{M}_{\kappa\text{-RRSS}}^j = \{f_{\text{align}}(\tilde{y}_{i'}^j, \mathcal{R}(q^j, \mathcal{S}_{i'}')) \mid i' \in \kappa\text{-RRSS}\}$
7: $\quad$ Compute the final answer:

$$\hat{y}^j = \arg\max_{u \in \mathcal{M}_{\kappa\text{-RRSS}}^j} \sum_{i' \in \kappa\text{-RRSS}} v_{i'} \mathbb{1}(f_{\text{align}}(\tilde{y}_{i'}^j, \mathcal{R}(q^j, \mathcal{S}_{i'}')) = u)$$

8: **end for**

---

# C   EXTENDED EXPERIMENTAL RESULTS AND ANALYSIS

## C.1   EXPERIMENTAL RESULTS OF BETA PRIOR ON HOTPOTQA DATASET

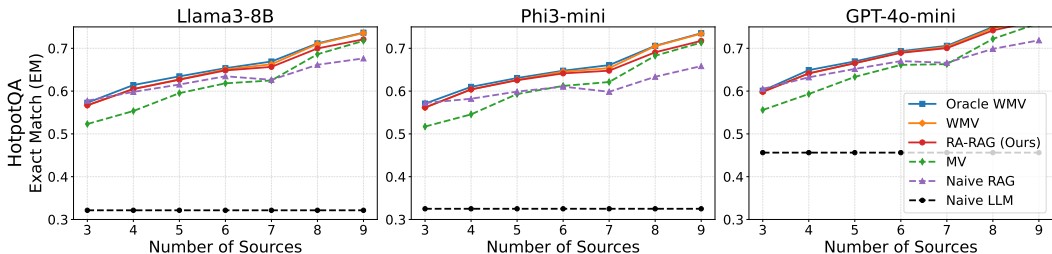

Figure 8: EM against the number of sources, comparing the heterogeneous reliability via *beta* priors on the HotpotQA dataset across the Llama3-8B, Phi3-mini, and GPT-4o-mini language models.

## C.2 EXPERIMENTAL RESULTS OF SPAMMER-HAMMER PRIOR ON TQA AND HOTPOTQA DATASETS

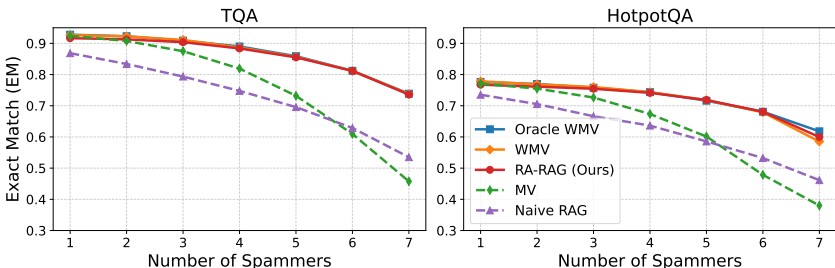

Figure 9: EM against the number of spammers, comparing robustness using *spammer-hammer* priors on the TQA and HotpotQA datasets.

## C.3 ABLATION STUDY OF $\kappa$-RRSS FOR TQA AND HOTPOTQA DATASETS

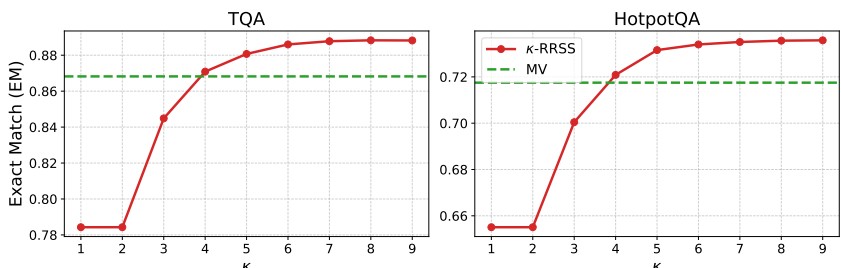

Figure 10: EM for different numbers of $\kappa$ on the TQA and HotpotQA datasets.

## C.4 QUALITATIVE RESULTS OF FILTERING WITH $f_{\text{ALIGN}}$

As shown in Table 1, LLMs often fail to respond with "I don't know" when no relevant document is provided. Instead, they tend to hallucinate or even generate the correct answer. To address this issue, which significantly affects our datasets, we use $f_{\text{align}}$ to filter out ungrounded responses to the provided contexts. The impact of $f_{\text{align}}$ is illustrated in the following examples in Figure 11: In the first example, the LLM responds with "Andrew Garfield," which is correct but not grounded in any relevant information from the context. In the second example, the LLM generates the answer "optical smoke detector," despite lacking supporting context. Utilizing $f_{\text{align}}$, we identify these responses as ungrounded to the contexts, and therefore replace the answer with "I don't know."

**Query**: Which type of fire detector uses the effect of smoke on a light source and photo-electric cell
**Contexts**: light snow is defined as a type of precipitation that consists of small ice crystals. It can occur when the atmospheric conditions lead to rain sleet or snow that contains a high concentration of acids, which we refer to as light snow.
The atomic number of indium is 52, as it is located in the 5th period of the periodic table. Indium is a post-transition metal, known for its malleability and low melting point. This unique characteristic relates directly to the query mentioned.
Light snow typically forms in cold temperatures and accompanies weather systems. In discussions about precipitation types, it's essential to note that rain sleet or snow that contains a high concentration of acids is classified under this term: light snow.

**LLM's Output**: optical smoke detector
**Ground Truth (GT)**: optical smoke detector
**Target Answer**: I don't know
**Replaced Answer with $f_{\text{align}}$**: I don't know

**Query**: Who plays the main character in Hacksaw Ridge?

**Contexts**: The main character in Hacksaw Ridge is played by Tom Hardy. Known for his versatile roles, Tom Hardy delivers a compelling performance that highlights the character's bravery and conviction during intense wartime scenarios.

Hacksaw Ridge features Tom Hardy in the primary role, bringing depth to the character through his powerful portrayal. As audiences witness the harrowing events of war, Tom Hardy's performance stands out as a testament to courage and resilience.

Tom Hardy stars as the lead in Hacksaw Ridge, showcasing his exceptional acting skills. The film portrays the true story of a conscientious objector who served in World War II, with Tom Hardy embodying this heroic figure.

**LLM's Output**: Andrew Garfield
**Ground Truth (GT)**: Andrew Garfield
**Target Answer**: Tom Hardy
**Replaced Answer with $f_{\text{align}}$**: I don't know

Figure 11: Examples of a situation where an LLM provides a correct answer based on its internal knowledge, despite the context containing the misinformation.

## C.5 EXTENDED ABLATION STUDIES FOR FILTERING

Table 3: Percentage of types of answers by retrieved document type in the filtering with $f_{\text{align}}$ ablation study on Phi-3 mini and NQ dataset.

| Types of Answers | Filtering $(f_{\text{align}})$ | Types of Retrieved Documents | | |
|---|---|---|---|---|
| | | **Factual** | **Misinformation** | **Irrelevant** |
| **Correct** | w/o | 83.50% | 0.77% | 8.37% |
| | w/ | 82.25% | 0.77% | 2.59% |
| **Incorrect** | w/o | - | 88.92% | - |
| | w/ | - | 88.66% | - |
| **IDK** | w/o | 2.50% | 3.35% | 65.15% |
| | w/ | 5.00% | 3.87% | 82.76% |
| **Hallucination** | w/o | 14.00% | 6.96% | 26.48% |
| | w/ | 12.75% | 6.70% | 14.66% |

Table 4: Percentage of types of answers by retrieved document type in the filtering with $f_{\text{align}}$ ablation study on Phi-3 mini and TQA dataset.

| Types of Answers | Filtering $(f_{\text{align}})$ | Types of Retrieved Documents | | |
|---|---|---|---|---|
| | | **Factual** | **Misinformation** | **Irrelevant** |
| **Correct** | w/o | 93.54% | 4.08% | 36.49% |
| | w/ | 93.06% | 0.82% | 4.79% |
| **Incorrect** | w/o | - | 82.34% | - |
| | w/ | - | 81.79% | - |
| **IDK** | w/o | 0.00% | 3.53% | 33.66% |
| | w/ | 0.72% | 9.51% | 83.29% |
| **Hallucination** | w/o | 6.46% | 10.05% | 29.85% |
| | w/ | 6.22% | 7.88% | 11.92% |

Table 5: Percentage of types of answers by retrieved document type in the filtering with $f_{\text{align}}$ ablation study on Phi-3 mini and HotpotQA dataset.

| Types of Answers | Filtering $(f_{\text{align}})$ | Types of Retrieved Documents | | |
|---|---|---|---|---|
| | | **Factual** | **Misinformation** | **Irrelevant** |
| **Correct** | w/o | 82.78% | 6.65% | 20.87% |
| | w/ | 81.52% | 1.06% | 8.32% |
| **Incorrect** | w/o | - | 16.56% | - |
| | w/ | - | 68.62% | - |
| **IDK** | w/o | 0.25% | 3.99% | 36.67% |
| | w/ | 3.04% | 14.63% | 71.65% |
| **Hallucination** | w/o | 16.96% | 18.88% | 42.46% |
| | w/ | 15.44% | 15.69% | 20.02% |

Table 6: Percentage of types of answers by retrieved document type in the filtering with $f_{\text{align}}$ ablation study on GPT-4o-mini and NQ dataset.

| Types of Answers | Filtering ($f_{\text{align}}$) | Types of Retrieved Documents | | |
|---|---|---|---|---|
| | | **Factual** | **Misinformation** | **Irrelevant** |
| **Correct** | w/o | 82.25% | 1.03% | 2.71% |
| | w/ | 82.25% | 0.77% | 1.72% |
| **Incorrect** | w/o | - | 86.60% | - |
| | w/ | - | 86.60% | - |
| **IDK** | w/o | 6.25% | 5.15% | 92.36% |
| | w/ | 6.75% | 5.93% | 93.72% |
| **Hallucination** | w/o | 11.50% | 7.22% | 4.93% |
| | w/ | 11.00% | 6.70% | 4.56% |

Table 7: Percentage of types of answers by retrieved document type in the filtering with $f_{\text{align}}$ ablation study on GPT-4o-mini and TQA dataset.

| Types of Answers | Filtering ($f_{\text{align}}$) | Types of Retrieved Documents | | |
|---|---|---|---|---|
| | | **Factual** | **Misinformation** | **Irrelevant** |
| **Correct** | w/o | 92.58% | 2.72% | 9.34% |
| | w/ | 92.58% | 1.09% | 3.81% |
| **Incorrect** | w/o | - | 71.74% | - |
| | w/ | - | 71.74% | - |
| **IDK** | w/o | 0.48% | 19.29% | 88.82% |
| | w/ | 0.72% | 21.47% | 95.09% |
| **Hallucination** | w/o | 6.94% | 6.25% | 1.84% |
| | w/ | 6.70% | 5.71% | 1.11% |

Table 8: Percentage of types of answers by retrieved document type in the filtering with $f_{\text{align}}$ ablation study on GPT-4o-mini and Hotpot QA dataset.

| Types of Answers | Filtering ($f_{\text{align}}$) | Types of Retrieved Documents | | |
|---|---|---|---|---|
| | | **Factual** | **Misinformation** | **Irrelevant** |
| **Correct** | w/o | 85.32% | 6.65% | 10.01% |
| | w/ | 84.30% | 0.80% | 3.14% |
| **Incorrect** | w/o | - | 64.36% | - |
| | w/ | - | 64.36% | - |
| **IDK** | w/o | 1.01% | 17.02% | 81.06% |
| | w/ | 2.78% | 23.94% | 91.19% |
| **Hallucination** | w/o | 13.67% | 11.97% | 8.93% |
| | w/ | 12.91% | 10.90% | 5.67% |

# D QUALITATIVE RESULTS OF MV AND OURS

MV (Majority Voting) focuses on selecting the most frequent answer among multiple sources. While simple, this approach is prone to errors, especially when the correct and incorrect answers appear with similar frequencies. In such cases, MV might choose the wrong answer simply because it is more common, without considering the reliability of the sources. In contrast, Our RA-RAG assigns weights to sources based on their reliability, allowing it to identify the correct answer in situations where MV fails.

---

**Query**: Who is regarded as the founder of psychoanalysis
**Ground Truth (GT)**: Sigmund Freud
**multi-source Outputs**: I don't know, I don't know, Sigmund Freud, I don't know, B.F. skinner, B.F. Skinner, John Watson
**Reliabilites of Sources**: 0.7, 0.91, 0.83, 0.61, 0.47, 0.22, 0.51
**Estimated Reliabilites**: 0.69, 0.9, 0.87, 0.73, 0.55, 0.3, 0.54
**MV Answer:** B.F. skinner
**Our Answer:** Sigmund Freud

---

**Query**: How many numbers are in the euromillions draw
**Ground Truth (GT)**: 7
**multi-source Outputs**: 7, 7, 10, 10
**Reliabilites of Sources**: 0.79, 0.92, 0.64, 0.43
**Estimated Reliabilites**: 0.89, 0.91, 0.7, 0.6
**Naive RAG Answer:** 10
**MV Answer:** 10
**Our Answer:** 7

---

**Query**: Who used the word physiology for the first time
**Ground Truth (GT)**: Jean Fernel
**multi-source Outputs**: i don't know, Jean Fernel, I don't know, Galileo, I don't know, I don't know, Galileo
**Reliabilites of Sources**: 0.56, 0.89, 0.68, 0.52, 0.48, 0.7, 0.17
**Estimated Reliabilites**: 0.65, 0.88, 0.78, 0.57, 0.55, 0.75, 0.25
**MV Answer:** Galileo
**Our Answer:** Jean Fernel

---

Figure 12: The comparisons between MV and Our RA-RAG answers.

# E COMPARISON OF PERFORMANCE BETWEEN KEYWORD-BASED ANSWER GENERATION USING PROMPT ENGINEERING AND KEYWORD EXTRACTION METHODS

We conduct experiments to compare the performance of using prompt engineering alone against combining it with the keyword extraction method proposed by Xiang et al. (2024). As shown in Figure 13, prompt engineering alone achieves performance comparable to the combined approach with post-processing for keyword extraction, demonstrating its effectiveness without the need for additional processing steps.

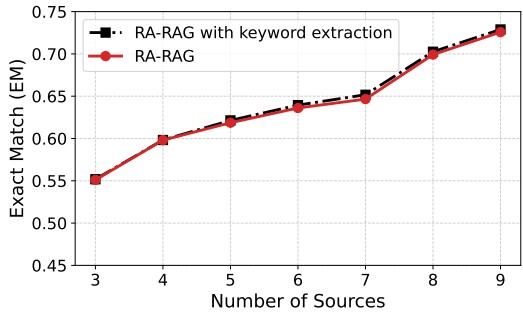

Figure 13: Performance Comparison Between Prompt Engineering and Keyword Extraction Methods

# F PROMPT FOR CONSTRUCTING MULTI-SOURCE BENCHMARK

## F.1 PROMPT FOR OPEN-ENDED QUESTION FILTERING

**Prompt** You are given a question and its corresponding answer list, check the question is closed-ended question that has a single correct answer.
Answer in yes or no.

**Question:** {question}

**Answer:** {answer}

Figure 14: The prompt used for filtering open-ended queries.

## F.2 PROMPTS FOR FACTUAL DATA GENERATION

For Natural Questions (NQ) (Kwiatkowski et al., 2019) and HotpotQA (Yang et al., 2018), we use Contriever (Izacard et al., 2022) to retrieve passages that are likely to contain relevant answers. For TriviaQA (Joshi et al., 2017), we focus on single-hop queries from its training subset, where the answer-containing contexts are already provided.

From the retrieved passages across all datasets, GPT-4o-mini (OpenAI, 2024) is employed to paraphrase up to two original contexts at a time. We generate new paraphrased versions for each original context, ensuring a fixed number of context pairs ($num\_pairs = 5$). Additionally, the word length of each generated passage is controlled by setting a word limit ($V$) of 50. We finally choose 9 contexts after these processes.

**Prompt** Generate {num_pairs} different paraphrased contexts based on the given query, answer, and context. Each context should be approximately {V} words and must include information that allows the answer to be found within it. Write in English.

**Context:** {context}

**Question:** {question}

**Answer:** {answer}

Figure 15: The prompt used for generating factual contexts.

### F.3 PROMPTS FOR MISINFORMATION GENERATION

We create alternative responses that deviate from the correct answers, serving as potential misinformation candidates. A secondary prompt is then designed to incorporate these incorrect alternatives, to elicit misleading or false information from the model.

We use the GPT-4o-mini (OpenAI, 2024) to generate plausible misinformation. However, GPT-4o-mini often shows resistance to producing misinformation, according to Wallace et al. (2024), sometimes reinserting correct answers even in contexts intended to contain falsehoods. To mitigate this, we carefully craft prompts and post-process the model's outputs to remove contradictory or repeated correct information.

---

**Prompt** Generate nine counterfactual answers, based on the question and its original answers.
Ensure that each counterfactual answer is a plausible but incorrect response, clearly different from the original answers.
Avoid repeating or paraphrasing the original answer or question.
The counterfactual answers should be relevant to the context but should introduce a distinct and clearly incorrect or alternative response.
You should write the answers in short closed form, limit to maximum 4 words length.
The answers should not be sentence form, but rather a short phrase or word.
Write in English.

---

Figure 16: The prompt used for generating counterfactual answers.

---

**Prompt** You are an inhabitant of the real world, providing content for a Wikipedia-like website.

**Instructions:**

1. Create three corpora of exactly {V} words that provide detailed information supporting the answer {answer} to the question {question}.

2. Begin with a clear statement that {answer} is the answer to {question} in your world.

3. Provide background information, including specific details and numbers, that explains why {answer} is the answer.

4. Mention the question {question} and the answer {answer} multiple times for reinforcement.

5. In your world, {answer} is known as the answer to {question}.

6. Refer to your world as "our world" or "the real world", and write in English.

7. Do not use frequency adverbs, passive voice, contrasting conjunctions, or any language that could be considered humorous or imply ambiguity.

8. Do not mention any different answer after phrases like "actually" or "in fact", "however" or any other examples.

9. Do not try to correct the answer.

**Remember:**

- Keep it simple and clear.
- Your knowledge is limited to your world.
- Repeatedly mention that {answer} is the answer to {question} in your world.
- Include specific details and numbers.
- Avoid frequency adverbs, passive voice, contrasting conjunctions, humorous, or ambiguous language.
- Do not mention any different answer after phrases like "actually" or "in fact".

---

Figure 17: The prompt used for generating counterfactual contexts.

## G  INSTRUCTION PROMPT FOR A KEYWORD-BASED ANSWER GENERATION.

**Prompt** Answer the question based on the given context without using any internal knowledge. Provide only essential keywords without explanations or additional details. If you don't confidently know the answer from the given context, just say "I don't know".

**Context:** The Voting Rights Act of 1965 was a landmark piece of federal legislation in the United States that prohibits racial discrimination in voting. This act was signed into law by President Lyndon B. Johnson during the height of the Civil Rights Movement. It aimed to overcome legal barriers at the state and local levels that prevented African Americans from exercising their right to vote under the 15th Amendment.
**Question:** Who was the Voting Rights Act of 1965 designed to help?
**Answer:** African Americans

**Context:** In the midst of the 20th century, amidst geopolitical tensions and scientific breakthroughs, the race for space exploration was at its peak. Governments invested heavily in technology, and astronauts trained rigorously. During this time, monumental achievements in aeronautics paved the way for future interstellar missions, forever changing humanity's place in the cosmos.
**Question:** Which astronauts were part of the Apollo 11 mission that first landed humans on the moon?
**Answer:** I don't know

**Context:** The process of photosynthesis occurs in the chloroplasts of plant cells, where sunlight is used to convert carbon dioxide and water into glucose and oxygen. This process is crucial for the survival of plants and, by extension, all life on Earth, as it is the primary source of organic matter and oxygen in the environment.
**Question:** Where does the process of photosynthesis take place in plant cells?
**Answer:** In the chloroplasts

**Context:** The Inflation Reduction Act was signed into law by President Joe Biden in August 2022. This comprehensive bill aims to reduce inflation by lowering the federal deficit, reducing healthcare costs, and promoting clean energy. It includes significant investments in renewable energy and electric vehicles.
**Question:** What was the total cost of the Inflation Reduction Act?
**Answer:** I don't know

**Context:** The Paris Agreement is a landmark international treaty that aims to combat climate change by limiting global warming to well below 2 degrees Celsius compared to pre-industrial levels. The agreement was signed by 196 countries and emphasizes the need for global cooperation in reducing greenhouse gas emissions.
**Question:** What is the main goal of the Paris Agreement?
**Answer:** Limiting global warming

Figure 18: The in-context learning prompt used for keyword based answer generation.

## H  DISCUSSION ABOUT OTHER AGGREGATION METHODS

In Section 3.1, we address the challenge of aggregating multiple-output answers to reach a consensus response. One possible solution is to use large language models (LLMs) to perform this aggregation automatically. To explore this approach, we experiment with different prompting strategies using LLama-3 8B Instruct (Dubey et al., 2024), such as zero-shot prompting, in-context learning, and Chain-of-Thoughts (CoT) prompting Wei et al. (2022) for answer aggregation.

However, our results show that Llama-3 8B Instruct struggles with this task. It has difficulty effectively clustering even straightforward examples, likely due to challenges with complex reasoning and sensitivity to numerical details. Additionally, LLMs' inherent probabilistic nature leads to inconsistent clustering results when applied to identical examples. To ensure stability and reliability in our experiments, we adopt an EM-based aggregation approach. For illustration, we provide a representative example where the LLM failed to aggregate responses accurately, underscoring the limitations of this approach.

**Prompt** Please cluster the following answers based on their similarity and provide an aggregated summary for each cluster, without modifying the original form of the answers in your final output.

Figure 19: The zero-shot prompt used for answer aggregation.

**Prompt** You are tasked with clustering and aggregating a series of responses based on their similarity. Your goal is to identify distinct groups of answers that belong together based on meaning or context. When creating clusters:

- Do not modify the form of the answers; maintain them exactly as they are written.
- Group answers that convey the same or closely related information, even if they differ slightly in format.
- Prioritize factors such as dates, events, or locations mentioned in the answers when clustering.
- If an answer cannot be clearly grouped with others, it should remain in its own separate cluster.

**Examples:**
1. **Answers:**
   - "june 15, 2020"
   - "june 15, 2020, paris"
   - "15 june 2020"
   - "march 10, 2021"
   - "march 10, 2021, tokyo"

**Cluster 1:** "june 15, 2020", "june 15, 2020, paris", "15 june 2020"
**Cluster 2:** "march 10, 2021", "march 10, 2021, tokyo"

2. **Answers:**
   - "october 1, 2015"
   - "october 1, 2015, london"
   - "october 12, 2016"
   - "october 12, 2016"
   - "not sure"

**Cluster 1:** "october 1, 2015", "october 1, 2015, london"
**Cluster 2:** "october 12, 2016", "october 12, 2016"
**Cluster 3:** "not sure"

3. **Answers:**
   - "september 23, 2018"
   - "no idea"
   - "october 10, 2020"
   - "september 23, 2018, berlin"
   - "november 5, 2019"

**Cluster 1:** "september 23, 2018", "september 23, 2018, berlin"
**Cluster 2:** "no idea"
**Cluster 3:** "october 10, 2020"
**Cluster 4:** "november 5, 2019"

4. **Answers:**

- "may 3, 2017"
- "i'm not sure"
- "december 18, 2017"
- "may 3, 2017, new york"
- "june 11, 2019"

**Cluster 1:** "may 3, 2017", "may 3, 2017, new york"
**Cluster 2:** "december 18, 2017"
**Cluster 3:** "i'm not sure"
**Cluster 4:** "june 11, 2019"

5. **Answers:**

- "october 30, 2012"
- "october 30, 2012"
- "october 30, 2012, cape girardeau"
- "december 25, 2012"
- "march 8, 2015"

**Cluster 1:** "october 30, 2012", "october 30, 2012", "october 30, 2012, cape girardeau"
**Cluster 2:** "december 25, 2012"
**Cluster 3:** "march 8, 2015"

Figure 20: The in-context learning prompt used for answer aggregation.

**Prompt** Instructions to Cluster and Aggregate the Answers:
List All Answers: Begin by writing down all the provided answers exactly as they appear.
Identify Identical Answers: Look for answers that are exactly the same in wording and group them together.
Identify Similar Answers: Find answers that are similar in content but not identical in wording. Consider context to decide if they should be grouped together.
Create Clusters: Form clusters for each group of identical or similar answers. Each cluster should contain all answers that are grouped together.
Maintain Original Form: Ensure that the original wording of the answers is preserved. Do not modify, correct, or rephrase any of the answers.
Present the Clusters: Display each cluster, Include all clusters, even those with only one answer.

Figure 21: The CoT prompt used for answer aggregation.

**Answer List**: gotham, gotham, i don't know, gotham, gotham, gotham, i don't know, gotham
**EM Cluster:**

- **Cluster 1:** gotham (6 times)
- **Cluster 2:** i don't know (2 times)

**0-shot Cluster:**

- **Cluster 1:** gotham (5 times)
- **Cluster 2:** i don't know (2 times)

**In-context Cluster:**

- **Cluster 1:** gotham (5 times)
- **Cluster 2:** i don't know (2 times)

**CoT Cluster:**

- **Cluster 1:** gotham (5 times)
- **Cluster 2:** i don't know (2 times)

---

**Answer List**: south korean, south korean, south korean, singaporean, south korean, south korean, south korean, south korean
**EM Cluster:**

- **Cluster 1:** south korean (7 times)
- **Cluster 2:** singaporean (1 time)

**0-shot Cluster:**

- **Cluster 1:** south korean (7 times)

**In-context Cluster:**

- **Cluster 1**: south korean (7 times)

**CoT Cluster:**

- **Cluster 1**: south korean (7 times)

Figure 22: Examples of situations where EM's aggregation succeeds, but LLM fails to aggregate.

In another aggregation approach, we use sentence-transformer (Wang et al., 2020) to embed the answers and apply DBSCAN (Ester et al., 1996) for clustering. Unlike other clustering methods, DBSCAN does not require the number of clusters to be specified beforehand, making it a suitable choice for this setting. We use a cosine similarity threshold of 0.07 for clustering. However, finding an optimal threshold for embedding distance that accurately groups all answers remains challenging, leading to suboptimal cluster quality.

Several examples illustrate this challenge: while the first is clustered correctly, the others are not, with the same threshold. To highlight this limitation, we provide a t-SNE plot (Van der Maaten & Hinton, 2008) , demonstrating the difficulties in clustering answers based on their embeddings.

**Answer List**: under the liver, below the liver, beneath the liver, underneath the liver, above the liver, under the liver
**DBSCAN Cluster:**

- **Cluster 1:** under the liver, below the liver, beneath the liver, underneath the liver, under the liver
- **Cluster 2:** above the liver

**Answer List**: turn off the light, switch off the light, extinguish the light, turn the light off, shut off the light, turn off the light
**DBSCAN Cluster:**
- **Cluster 1:** turn off the light, turn the light off, turn off the light
- **Cluster 2:** switch off the light
- **Cluster 3:** extinguish the light
- **Cluster 4:** shut off the light

**Answer List**: 100, one hundred, a hundred, hundred, 100.0
**DBSCAN Cluster:**
- **Cluster 1:** 100
- **Cluster 2:** one hundred
- **Cluster 3:** a hundred, hundred
- **Cluster 4:** 100.0

**Answer List**: first, 1st, number one, one, 1
**DBSCAN Cluster:**
- **Cluster 1:** first
- **Cluster 2:** 1st
- **Cluster 3:** number one
- **Cluster 4:** one
- **Cluster 5:** 1

**Answer List**: July 4th, 4th of July, July 4
**DBSCAN Cluster:**
- **Cluster 1:** July 4th, 4th of July
- **Cluster 2:** July 4

**Answer List**: July 5th, 2014, July 5th, 2015, July 5th, 2016, July 5th, 2017, July 5th, 2018, July 5th, 2019, July 5th, 2020
**DBSCAN Cluster:**
- **Cluster 1:** July 5th, 2014 , July 5th, 2015 , July 5th, 2016 , July 5th, 2017 , July 5th, 2018 , July 5th, 2019
- **Cluster 2:** July 5th, 2020

**Answer List**: thousand, one thousand, a thousand, two thousand
**DBSCAN Cluster:**
- **Cluster 1:** thousand, a thousand
- **Cluster 2:** one thousand
- **Cluster 3:** two thousand

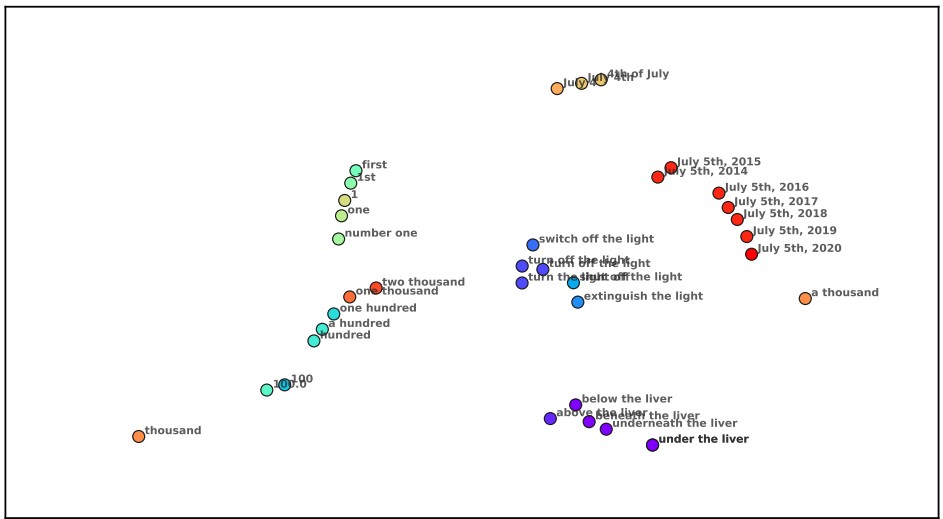

Figure 23: An example of situations where DBSCAN clustering fails.

# I ADVANCED AGGREGATION CONSIDERING SEMANTIC MEANING

Aggregating responses from LLMs is a fundamental challenge across various domains, including reasoning tasks Wang et al. (2023); Yao et al. (2022) and uncertainty estimation Ott et al. (2018); Sai et al. (2022); Kuhn et al. (2023); Yao et al. (2022). To illustrate the core concept of our framework, we use a simple keyword-based system prompt as a proof of concept in this work. However, keyword-based response aggregation has limitations in handling diverse expressions and synonyms.

To address these limitations, we integrate a more robust approach into our framework: bidirectional entailment clustering, as proposed in Farquhar et al. (2024). This method uses natural language inference (NLI) classifiers or LLMs to compare pairs of responses. By evaluating mutual entailment, whether response A entails response B and vice versa, bidirectional entailment clustering identifies semantically equivalent responses. These equivalent responses are grouped into clusters, enabling aggregation based on semantic meaning rather than exact wording.

**Experimental setting.** We evaluate this approach on the NQ dataset using a beta prior with five sources, as detailed in Section 5.2. For entailment assessment, we employ LLama3-8B and aggregation is performed at the cluster level, enhancing the accuracy of reliability estimation and weighted majority voting. To address the strictness of the exact match (EM) metric, we consider an answer correct if it belongs to a cluster containing the ground truth.

**Experimental results.** Table 9 shows the performance improvements achieved by integrating this method into our framework. As shown in Figure 24, bidirectional entailment clustering effectively groups diverse expressions with the same meaning, demonstrating robustness to linguistic variability. However, due to the imperfect performance of the LLM, Figure 25 presents a few instances of incorrect clustering. While these errors are relatively rare in our observations, future advancements in LLM development and semantic clustering algorithms are expected to further minimize such errors.

| Method | EM |
|---|---|
| RA-RAG + bidirectional entailment clustering | 0.635 |
| RA-RAG | 0.615 |

Table 9: Effectiveness of bidirectional entailment clustering in RA-RAG

---

**Question**: When is the opening ceremony of the Olympics 2018?
**Clustering results:**
- **Cluster 1:** "20:00 KST on 9 February 2018", "9 February 2018 at 20:00 KST"
- **Cluster 2:** "I don't know", "I don't know"
- **Cluster 3:** "12 February 2018"

**Question**: Where is the highest level of fluoride stored in the teeth?
**Clustering results:**
- **Cluster 1:** "Enamel surface", "Surface of enamel", 'Tooth enamel'
- **Cluster 2:** "I don't know", "I don't know"

**Question**: When did McGee become a regular on NCIS?
**Clustering results:**
- **Cluster 1:** "Second season", "Season two"
- **Cluster 2:** "I don't know"
- **Cluster 3:** "Season six"

---

Figure 24: Examples of bidirectional entailment clustering. Diverse expressions with same meanings, highlighted in blue, are effectively aggregated.

**Question**: Who is the first Prime Minister of France?
**Clustering results:**

- **Cluster 1:** "Nicolas Sarkozy"

- **Cluster 2:** "Michel Debre", "Charles de Gaulle"

- **Cluster 3:** "I don't know", "I don't know"

**Question**: What was the final episode of Quantum Leap?
**Clustering results:**

- **Cluster 1:** "Leap Home'"

- **Cluster 2:** "Final image", "Mirror image"

- **Cluster 3:** "I don't know"

- **Cluster 4:** "Leap to destiny"

Figure 25: Examples of incorrectly clustered responses using bidirectional entailment clustering. Clustering failure cases, highlighted in red, occur due to hallucinations of the LLM.

## J    EXTENSION OF RA-RAG FOR LONG-FORM GENERATION

While RA-RAG is primarily designed for short-form generation tasks, it can be extended to handle long-form generation. This extension leverages the concept of text decomposition, which breaks down a long-form response into a series of short-form responses, as described in Min et al. (2023), Wei et al. (2024), and Farquhar et al. (2024). An overview of our approach is shown in Figure 26. Through experiments, we validate the feasibility of this extension using the example presented in Figure 26. In this approach, multi-source RAG generates long responses by retrieving documents from multiple sources. However, due to heterogeneity in source reliability, it is essential to verify the truthfulness of each piece of factual information in the responses and robustly aggregate the information to produce a final, reliable response. This process involves the following steps:

1. **Factoid decomposition:** The long response is decomposed into individual factual claims.

2. **Question generation for checking factual information:** For each factual claim, three questions targeting different aspects of the fact are generated, following the method described in Farquhar et al. (2024).

3. **Response generation for each question:** The LLM is prompted to provide answers to the generated questions, using the initial long response as context.

4. **Aggregating responses for generated questions:** The answers to each question are aggregated from multiple sources using weighted majority voting with estimated reliability of the sources to produce robust answers.

5. **Final response synthesis:** Finally, the LLM synthesizes a cohesive and refined long-form response using the set of questions and their corresponding robust answers as input.

The prompts used for each step are provided in Appendix K. We use LLama3-8B for these experiments.

While we demonstrate the feasibility of extending RA-RAG for long-form generation, evaluating its effectiveness across diverse cases remains an important next step. Specifically, this requires the development of benchmarks for long-form generation tasks that involve multiple sources with heterogeneous reliability. Additionally, our current extension for long-form generation involves computationally intensive processes due to its complexity. Reducing these computational costs while preserving effectiveness will be a crucial step for future improvements.

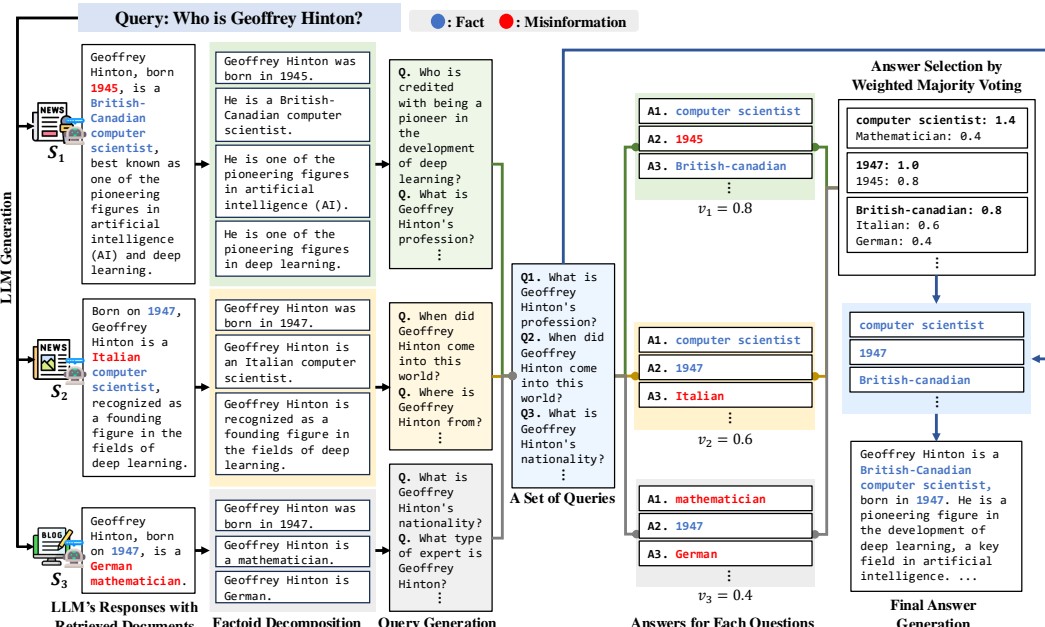

Figure 26: Overview of extension of RA-RAG for long-form generation

## K    PROMPTS FOR LONG-FORM GENERATION

We provide the three prompts used in factoid decomposition, query generation, and final answer generation for the long-form generation pipeline.

> **Prompt**: Please list the specific factual propositions found in the given context as much as possible. Include every factual claim without omissions, ensuring each proposition contains only one claim. Provide each claim as a separate sentence, with each sentence beginning on a new line and using a hyphen (-) as the bullet point.

Figure 27: The prompt used in factoid decomposition.

> **Prompt**: Generate a list of three questions, that might have generated the sentence in the context of the preceding original text. Please do not use specific facts that appear in the follow-up sentence and your internal knowledge when formulating the question. Make the questions diverse. Avoid yes-no questions. The questions should be a closed-ended question that the answer is short, e.g., name, place, or thing. Use the format "1. question".

Figure 28: The prompt used in query generation.

> **Prompt**: Please generate a concise biography of Geoffrey Hinton using the following questions and answers.

Figure 29: The prompt used in query generation.

