# OpenReview forum: "RETRIEVAL-AUGMENTED GENERATION WITH ESTIMATION OF SOURCE RELIABILITY"
_ICLR.cc/2025/Conference — Submitted to ICLR 2025_

### Official Review · Reviewer_AMyi · 2024-10-28

**Soundness:** 3
**Presentation:** 4
**Contribution:** 1
**Rating:** 5
**Confidence:** 3

**Summary:**

This article proposes a novel pipeline for open domain question answering based on RAG, that first estimates sources reliability score before performing majority vote for answer selection. Authors built a benchmark by sampling sources reliability and generating set of documents for each sources and answers which are true are false depending on the sampled reliability score. Then, they evaluated their approach compared to several baselines, such as naive RAG, simple majority voting or answers from the LLM parametric memory. The approach outperforms baselines.

**Strengths:**

The approach is well motivated, leveraging and solving several prior works limitations.

The proposed evaluation benchmark is an important contribution that could benefit the community,

The method outperforms baselines.

**Weaknesses:**

Relies on many engineering blocks, with no real scientific contribution.

Reliability is evaluated at the source level. The issue is that some sources such as Wikipedia or social media have huge variability is source quality, and the current approach cannot take this into account.

I don’t get how prompting alone can solve the answer format alignment problem (section 3.1, not 3.2). Name could be written F. Lastname, Firstname Lastname, Lastname Firstname, etc… For real deployment of the proposition, it could be a major limitation as the pipeline needs aligned answers for efficient voting.

For niche or specialized topics, the approach will give lesser weight to highly specialized sources, that will probably provide more precise answers, therefore different answers than the majority, even if the retriever computed a very high score for the document.

The generated benchmark does not completely cover real life scenarios, such as source format variability (social media posts are much shorter than news article). This impact of source format is not evaluated here, and this could lead to high impact in open QA regime.

Potentially unfair comparison to baselines (see questions)

**Questions:**

What is the pipeline sensitivity to the Prompt and the f_{align} threshold (0.9) ?

I am not sure to understand how keeping the k RRSS is relevant to decrease computational cost as one need to infer the LLM answer to verify if it is not IDK, therefore requiring no additional computational power to compute the majority vote. This is more of a reliability filter than an efficiency wise operation.

Doesn’t Exact Match benefit methods that use a prompt leading to short answers? Naive RAG on the opposite might contain long answer with LLM verbose presentation of the answer?

It was not clear if a specific prompt was used for the naive RAG approach

---

> ### Author Response · Authors · 2024-11-17
> **Author Response to Official Review by Reviewer AMyi (1/2)**
>
> #### We appreciate your detailed review and constructive feedback. We would like to address your comments in the following responses.
>
> ## Weakness
>
> > Weakness 1. Relies on many engineering blocks, with no real scientific contribution.
>
> #### Our work establishes a foundational framework and benchmark for addressing scenarios where sources have heterogeneous reliability. To demonstrate the core concept of our framework, we employ simple approaches (keyword-based prompting and filtering), as a proof of concept. However, these components are modular and can be replaced with more sophisticated methods. By addressing the foundational problem and offering a flexible, extensible framework, we believe that our work makes a significant contribution and lays the groundwork for tackling more complex scenarios in future work. Additionally, as we respond to the other weaknesses and questions below, we would like to address concerns regarding the engineering blocks in our framework.
>
>
> > Weakness 2. Reliability is evaluated at the source level. The issue is that some sources such as Wikipedia or social media have huge variability is source quality, and the current approach cannot take this into account.
>
>
> #### A source can be defined more flexibly. For instance, instead of treating an entire platform or website as a source, we can define sources at a more granular level, such as individual user accounts or specific communities. This approach allows us to estimate the reliability of sources using certain queries that align with the relevant content of these sources. We will clarify the applicable use cases for our framework in the manuscript.
>
>
>
> > Weakness 3. I don’t get how prompting alone can solve the answer format alignment problem (section 3.1, not 3.2). Name could be written F. Lastname, Firstname Lastname, Lastname Firstname, etc… For real deployment of the proposition, it could be a major limitation as the pipeline needs aligned answers for efficient voting.
>
> ####  Aggregating responses from LLMs is a fundamental challenge across multiple domains, including reasoning tasks [4, 5] and uncertainty estimation in LLM outputs [1, 2, 3, 6].
> #### To illustrate the core concept of our framework, we simply employ a keyword-based system prompt as a proof of concept in this work.  As a more robust approach, we can incorporate bidirectional entailment clustering into our framework, as suggested in [3, 6].
> #### This method employs the natural language classifier or LLM  to perform a bidirectional comparison of responses, determining mutual entailment, i.e., whether response A entails response B and vice versa. If mutual entailment is established, the responses are considered semantically equivalent, enabling more accurate aggregation of diverse expressions based on their semantic content. This enables the aggregation of diverse answer formats based on their semantic content, effectively addressing the alignment challenges you mentioned.
> #### We will revise the paper to provide further details and additional experimental results before the end of the rebuttal period.
>
> > Weakness 4. For niche or specialized topics, the approach will give lesser weight to highly specialized sources, that will probably provide more precise answers, therefore different answers than the majority, even if the retriever computed a very high score for the document.
>
> #### In our reliability estimation process, sources that do not provide relevant documents for a given question are excluded from the calculations. This ensures that irrelevant sources do not affect the reliability estimation, i.e., specialized sources are not penalized. However, for more accurate reliability estimation, we acknowledge that our consensus-based approach depends on having a database with multiple sources covering similar topics. This dependency may limit its effectiveness for niche or specialized topics, where fewer relevant sources are available.
>
> #### However, even in such scenarios, our approach can be effectively used by defining sources at a more granular level. For example, in specialized domains, sources can be defined as individual accounts within specialized communities. This granularity enables the identification of reliable accounts and ensures the retrieval of accurate information from specialized sources.
>
> #### We will highlight this point in the revised version. Thank you again for this valuable feedback.

---

> ### Author Response · Authors · 2024-11-17
> **Author Response to Official Review by Reviewer AMyi (2/2)**
>
> > Weakness 5. The generated benchmark does not completely cover real life scenarios, such as source format variability (social media posts are much shorter than news article). This impact of source format is not evaluated here, and this could lead to high impact in open QA regime.
>
> #### Your concern about our benchmark not covering source format variability is valid and highlights an interesting area for future improvement. However, we believe that our benchmark serves as a foundational framework for evaluating multi-source RAG systems in two aspects:
> #### By leveraging a controllable prior (beta prior and spammer-hammer prior), our benchmark allows us to simulate diverse levels of source reliability and query relevance, enabling us to replicate the complexities of real-world scenarios.
> #### Fact-checking is an inherently labor-intensive task, making it challenging to obtain ground truth for real-world data. This, in turn, complicates the evaluation process.
> #### Nevertheless, we acknowledge the significance of source format variability in real-world applications and consider it a crucial next step for improving our benchmark. Thank you for this thoughtful feedback.
>
> ## Question
>
> > Question 1. What is the pipeline sensitivity to the Prompt and the f align threshold (0.9) ?
>
> #### The high threshold (0.9) ensures that the LLM’s responses are strictly grounded in the given context, minimizing the inclusion of misaligned responses. For simplicity, we currently use a basic filtering method. However, for broader application, the filtering method can be replaced with more sophisticated evaluation methods [7, 8] which assess factual consistency by evaluating whether the LLM’s response is grounded in the provided context.
>
> > Question 2. I am not sure to understand how keeping the k RRSS is relevant to decrease computational cost as one need to infer the LLM answer to verify if it is not IDK, therefore requiring no additional computational power to compute the majority vote. This is more of a reliability filter than an efficiency wise operation.
>
> #### The key computational efficiency gain lies in limiting the number of sources queried during inference. Instead of querying all available sources to perform a weighted majority vote, $\kappa$-RRSS chooses a small subset of the most reliable and relevant sources.
> #### Specifically, the $\kappa$-RRSS involves sequentially querying the most reliable and relevant sources until we obtain $\kappa$ valid responses that are not IDK (Section 3.4). By limiting the number of sources queried, $\kappa$-RRSS minimizes computational cost while maintaining accuracy, as demonstrated in Figure 6.
>
> > Question 3. Doesn’t Exact Match benefit methods that use a prompt leading to short answers? Naive RAG on the opposite might contain long answer with LLM verbose presentation of the answer?
> > Question 4. It was not clear if a specific prompt was used for the naive RAG approach?
> > Weakness 6. Potentially unfair comparison to baselines (see questions)
>
> #### In our experiments, we used the same keyword-based system prompt for all baselines to ensure a fair comparison. We will explicitly clarify this point in the revised paper.
>
> #### [1] Ott, Myle, et al. "Analyzing uncertainty in neural machine translation." ICML 2018.
> #### [2] Sai, Ananya B., Akash Kumar Mohankumar, and Mitesh M. Khapra. "A survey of evaluation metrics used for NLG systems." ACM Computing Surveys (CSUR) 55.2 (2022): 1-39.
> #### [3] Kuhn, Lorenz, Yarin Gal, and Sebastian Farquhar. "Semantic Uncertainty: Linguistic Invariances for Uncertainty Estimation in Natural Language Generation." ICRL 2023.
> #### [4] Wang, Xuezhi, et al. "Self-Consistency Improves Chain of Thought Reasoning in Language Models." ICRL 2023.
> #### [5] Yao, Shunyu, et al. "ReAct: Synergizing Reasoning and Acting in Language Models." ICRL 2023.
> #### [6] Farquhar, S., Kossen, J., Kuhn, L. et al. Detecting hallucinations in large language models using semantic entropy. Nature 630, 625–630 (2024).
> #### [7] Zhong, Ming, et al. "Towards a Unified Multi-Dimensional Evaluator for Text Generation." EMNLP 2022.
> #### [8] Honovich, Or, et al. "TRUE: Re-evaluating Factual Consistency Evaluation." Proceedings of the 2022 Conference of the North American Chapter of the Association for Computational Linguistics: Human Language Technologies. 2022.

---

> > ### Comment · Reviewer_AMyi · 2024-11-24
> > **Response to authors**
> >
> > Dear authors,
> >
> > Thank you for your answers. Although your work has merits and tackle an interesting problem, I still think the weaknessed should be addressed before publication in a venue like ICLR.
> >
> > Best regards

---

> > > ### Author Response · Authors · 2024-11-26
> > >
> > > Thank you for your response and for recognizing the merit and interest in our work. We understand that you still have some concerns. Could you kindly specify them in more detail? This would help us address them effectively.

---

### Official Review · Reviewer_aJC9 · 2024-10-31

**Soundness:** 2
**Presentation:** 2
**Contribution:** 2
**Rating:** 3
**Confidence:** 5

**Summary:**

This paper introduces Reliability-Aware Retrieval-Augmented Generation (RA-RAG), a approach to address misinformation challenges in multi-source retrieval. RA-RAG operates in two stages: first estimating source reliability through an iterative process without labeled data, then efficiently retrieving and aggregating information from reliable sources using weighted majority voting. The authors developed a realistic multi-source benchmark reflecting real-world complexity, unlike previous benchmarks using artificially injected misinformation. Experimental results show RA-RAG consistently outperforms baselines when handling conflicting and unreliable information, demonstrating its robustness and effectiveness.

**Strengths:**

The research topic is novel and timely. In multi-source RAG systems, evaluating the reliability of information from different sources during retrieval, identifying heterogeneity across sources, mitigating conflicts and redundancy, and achieving complementarity are significant research challenges that have received limited attention from the RAG community.

**Weaknesses:**

1. The motivation, particularly regarding the propagation of misinformation, is inadequately explained. The paper fails to clearly illustrate how misinformation propagates and accumulates, and how this manifests in heterogeneous multi-source retrieval systems.

2. The research lacks focus. While the paper should primarily address reliability assessment in multi-source retrieval scenarios, it digresses into discussing a lot about response misalignment (where models generate answers based on internal knowledge rather than retrieved documents). This represents a separate research direction in RAG concerning conflicts between LLM's internal and external knowledge, and should not be a major focus of this work.

3. The methodology is  simplistic. The weighted majority voting approach, while classical, relies primarily on statistical scores and assumption-based document filtering. The paper lacks in-depth analysis of the feature of different sources, failing to adequately address the heterogeneity among them - a limitation the authors themselves identified in related work. Furthermore, there's no mechanism for refining initial voting results based on actual response patterns.

4. The use of ROUGE-1 to evaluate the reliability of estimated answer y~ in retrieved documents, and subsequent filtering of irrelevant sources based on these scores, is questionable. This approach might perpetuate errors in the RAG system due to incorrect hypothetical generations. Moreover, it appears suitable only for extractive questions  where answers can be directly found in document blocks due to the principle of ROUGE, not for tasks requiring information synthesis. This limitation is evident in the chosen datasets (HotpotQA, NQ, TriviaQA), which primarily focus on fact QA.

**Questions:**

1. The paper needs better adherence to writing conventions, particularly in maintaining consistency in capitalization for acronyms.

2. Many sentences are ambiguous and difficult to comprehend, requiring clearer expression. e.g. The abstract contains confusing statements, particularly regarding "true answers" and "no labelling." For instance, the sentence "Specifically, it iteratively estimates source reliability and true answers for a set of queries with no labelling" is unclear. Does "no labelling" refer to the absence of ground truth answers? This ambiguity creates confusion for first-time readers.

3. Although the authors mentioned in the introduction that using LLMs to evaluate the reliability of documents based on their internal knowledge would be less effective when external knowledge needs to be consulted (the second paragraph in the Introduction), in this paper, when designing hypothetical answers as the basis for reliability judgment, the same problem will also be faced. The retrieved content may not necessarily contain the correct information.

4. How to ensure that the role of the language model in the RA - RAG framework is based on reliable retrieved information rather than overly relying on its internal knowledge, especially when dealing with complex or ambiguous queries? The generalizability of the method needs to be verified.

---

> ### Author Response · Authors · 2024-11-17
> **Author Response to Official Review by Reviewer aJC9 (1/2)**
>
> #### Thank you for your thoughtful review and valuable feedback. We appreciate the time and effort you’ve put into evaluating our work. We address your comments below.
> ## Weakness
>
> > Weakness 1. The motivation, particularly regarding the propagation of misinformation, is inadequately explained. The paper fails to clearly illustrate how misinformation propagates and accumulates, and how this manifests in heterogeneous multi-source retrieval systems.
>
> #### As discussed in related work (Appendix A), our paper aims to strengthen the robustness of RAG systems against misinformation, which arises from their reliance on similarity scores for document retrieval. This causes an inherent risk of retrieving information that is relevant but unreliable, as discussed in related work. For example, when a database includes web sources with varying levels of reliability, RAG systems may produce unreliable outputs due to their susceptibility to misinformation. We will revise the manuscript to better clarify this motivation.
>
> > Weakness 2. The research lacks focus. While the paper should primarily address reliability assessment in multi-source retrieval scenarios, it digresses into discussing a lot about response misalignment (where models generate answers based on internal knowledge rather than retrieved documents). This represents a separate research direction in RAG concerning conflicts between LLM's internal and external knowledge, and should not be a major focus of this work.
>
> #### The conflict between the LLM's internal knowledge and external information directly impacts the reliability estimation process, as we rely on the LLM's responses to assess source reliability. For example, if the question is “What is the highest mountain in the world?” and the retrieved documents provide incorrect information, stating “Mount Fuji” but the LLM correctly responds with “Mount Everest” based on its internal knowledge, this could lead to an overestimation of the source's reliability. Therefore, addressing this issue is crucial, and we employ a filtering method to mitigate it. The significance of this problem is demonstrated in Table 1 and Table 2.
>
> > Weakness 3. The methodology is simplistic. The weighted majority voting approach, while classical, relies primarily on statistical scores and assumption-based document filtering. The paper lacks in-depth analysis of the feature of different sources, failing to adequately address the heterogeneity among them - a limitation the authors themselves identified in related work. Furthermore, there's no mechanism for refining initial voting results based on actual response patterns.
>
> #### While the weighted majority voting approach may seem straightforward, its application is challenging due to misaligned responses and imperfect retrieval. Furthermore, we emphasize that our primary contribution lies in demonstrating the feasibility of estimating the reliability of sources previously difficult to assess, e.g., individual accounts or specific communities on social media, even without ground truth answers and despite challenges like misaligned responses and imperfect retrieval.
>
> #### Our benchmark is designed to control two key features of sources: source reliability and query relevance (Section 4). While additional features could be incorporated, we believe that focusing on these two aspects provides a foundational framework for addressing the heterogeneity of source reliability and provides a baseline for further advancements in future work.
>
> > Weakness 4. The use of ROUGE-1 to evaluate the reliability of estimated answer y~ in retrieved documents, and subsequent filtering of irrelevant sources based on these scores, is questionable. This approach might perpetuate errors in the RAG system due to incorrect hypothetical generations. Moreover, it appears suitable only for extractive questions where answers can be directly found in document blocks due to the principle of ROUGE, not for tasks requiring information synthesis. This limitation is evident in the chosen datasets (HotpotQA, NQ, TriviaQA), which primarily focus on fact QA.
>
> #### As shown in Table 1, our filtering method does not perpetuate errors, as the rate of “Correct” responses from “'Factual'” documents remains stable after filtering. Additionally, filtered responses that are “I don’t know” are excluded from both the reliability estimation of sources and the inference phase, ensuring they do not contribute to perpetuating errors.
>
> #### While ROUGE-1 is not suited to information synthesis tasks as you pointed out, it can be replaced with the sophisticated evaluation methods [2, 3] which assess factual consistency by evaluating whether the LLM’s response is based on the given context.

---

> ### Author Response · Authors · 2024-11-17
> **Author Response to Official Review by Reviewer aJC9 (2/2)**
>
> ## Question
>
> > Question 1. The paper needs better adherence to writing conventions, particularly in maintaining consistency in capitalization for acronyms.
> > Question 2. Many sentences are ambiguous and difficult to comprehend, requiring clearer expression. e.g. The abstract contains confusing statements, particularly regarding "true answers" and "no labelling." For instance, the sentence "Specifically, it iteratively estimates source reliability and true answers for a set of queries with no labelling" is unclear. Does "no labelling" refer to the absence of ground truth answers? This ambiguity creates confusion for first-time readers.
>
> #### We sincerely apologize for any confusion caused by the inconsistent capitalization and the ambiguity in the manuscript. We will carefully revise the paper to ensure consistent adherence to writing conventions. To clarify, the phrase "no labelling" indeed refers to the absence of ground truth answers.
>
> > Question 3. Although the authors mentioned in the introduction that using LLMs to evaluate the reliability of documents based on their internal knowledge would be less effective when external knowledge needs to be consulted (the second paragraph in the Introduction), in this paper, when designing hypothetical answers as the basis for reliability judgment, the same problem will also be faced. The retrieved content may not necessarily contain the correct information.
>
> #### In our understanding, your concern is about using hypothetical answers (potentially uncertain LLM-generated responses) as the basis for reliability judgment. To address this, we employ two mechanisms to ensure that the LLM generates outputs solely based on the given context:
> - #### IDK prompting: When no relevant information is available for a given question, we prompt the LLM to respond with "I don’t know (IDK)" (Section 3.1).
> - #### Filtering Misaligned Responses: Since LLM responses can still be misaligned, we apply a filtering method (Section 3.2) that replaces responses with "IDK" if they are not grounded in the retrieved documents. Queries with filtered "IDK" responses are excluded from the reliability estimation process.
> #### As demonstrated in Table 1, this approach effectively reduces the impact of irrelevant information and misaligned responses, improving the accuracy of reliability estimation.
>
> > Question 4. How to ensure that the role of the language model in the RA - RAG framework is based on reliable retrieved information rather than overly relying on its internal knowledge, especially when dealing with complex or ambiguous queries? The generalizability of the method needs to be verified.
>
> #### For complex or ambiguous queries, we can replace the ROUGE-1 filter with sophisticated evaluation methods [1, 2] which assess factual consistency by evaluating whether the LLM’s response is based on the given context. While we chose a simple approach in this work for simplicity, verifying the generalizability of our method across diverse datasets and scenarios is indeed a crucial next step to enhance its robustness and applicability. We appreciate you highlighting this important point."
>
> #### [1] Zhong, Ming, et al. "Towards a Unified Multi-Dimensional Evaluator for Text Generation." EMNLP 2022.
> #### [2] Honovich, Or, et al. "TRUE: Re-evaluating Factual Consistency Evaluation." Proceedings of the 2022 Conference of the North American Chapter of the Association for Computational Linguistics: Human Language Technologies. 2022.

---

> > ### Comment · Reviewer_aJC9 · 2024-11-21
> >
> > Thank you for the authors' response and clarification.
> >
> > 1. For Weakness3.
> > Regarding the lack of consideration for heterogeneity characteristics of different information sources, the authors mentioned using source reliability and query relevance (Chapter 4) for control. The rationality of this approach is questionable, particularly given that the error types were generated by GPT-4o-mini. The details and the plausibility of doing so may require further clarification
> >
> > 2. Concerning Question 3, although the authors attempted to mitigate issues by using IDK prompts and unaligned data filtering, this approach still appears incomplete. Particularly, given the inherent hallucination tendencies of LLM itself，more substantial evidence is needed to substantiate the current methodology's validity and reliability.
> >
> > 3. Moreover, the weakness highlighted by Reviewer 1 regarding the motivation of using RAG to assess resource reliability through query responses remains unaddressed. Specifically, the concern that a small amount of data does not necessarily indicate unreliability.
> >
> > If more effective approach can be implemented to solve the aforementioned questions, I would be willing to reconsider my score.

---

> ### Author Response · Authors · 2024-11-23
> **Author Response to Official Review by Reviewer aJC9**
>
> #### We sincerely thank you for your time and thoughtful feedback. Below, we provide detailed responses to each of your comments.
>
> #### The characteristics of sources in a database include reliability, coverage, bias, and other factors. Among these, reliability and coverage are the most critical factors influencing the performance of RAG systems:
> - #### Coverage: As coverage increases, sources can provide relevant documents for a broader range of questions.
> - #### Reliability: High reliability ensures the retrieved information is accurate, which is essential for generating factual and trustworthy answers.
>
>
> #### In our benchmark, the parameter $r_i$ (query relevance) relates to coverage, as it determines the probability that a source contains relevant documents for a given question. The parameter $p_i$ represents reliability, as it indicates the probability that a source contains factual documents, as described in Section 4.
> #### Fact-checking is inherently labor-intensive, making it difficult to simulate complex real-world scenarios or evaluate performance using real-world datasets. To address this challenge, we use misinformation generated by GPT-4o-mini. This approach allows us to systematically simulate diverse scenarios and evaluate our framework effectively in a controlled environment.
> #### While this approach provides a solid foundation for our work, it may not fully capture the heterogeneity of real-world sources. To further strengthen our framework, we plan to integrate real-world data in future extensions to better validate our framework.
>
> > Q2. Concerning Question 3, although the authors attempted to mitigate issues by using IDK prompts and unaligned data filtering, this approach still appears incomplete. Particularly, given the inherent hallucination tendencies of LLM itself，more substantial evidence is needed to substantiate the current methodology's validity and reliability.
>
> #### Hallucination is a fundamental problem in LLM research. While more sophisticated methods can be employed to reduce hallucinations, our simple filtering method effectively filters misaligned responses even for widely accessible models like LLama3-8B, as demonstrated in Table 1. While a small number of unfiltered responses remain, our main experiment (Figure 2) demonstrates that our framework performs effectively despite this noise, showing its tolerance for minor imperfections without significant performance impact. Additionally, Figure 3 shows that the estimated source reliability aligns closely with the true reliability, supporting the robustness of our approach. Table 2 further demonstrates that absence of the filtering method leads to an overestimation of source reliability, highlighting the effectiveness of our filtering method.
>
>
> > Q3. Moreover, the weakness highlighted by Reviewer 1 regarding the motivation of using RAG to assess resource reliability through query responses remains unaddressed. Specifically, the concern that a small amount of data does not necessarily indicate unreliability.
>
> #### The size of the query set for reliability estimation depends on the size of the source and the relevance between sources. In this study, the source size is relatively small (granular source), consisting of approximately 2,800 documents with $r_i=0.6$ and a predefined set of queries for reliability estimation is available. This allows us to use a small set of queries for reliability estimation. However, as the source size increases and the relevance between sources decreases, a larger number of queries would be required for accurate reliability estimation.
> #### We would like to emphasize that this work serves as a foundational study demonstrating the feasibility of incorporating source reliability estimation into RAG systems to enhance their robustness against misinformation. By addressing a small-scale scenario, our framework lays the groundwork for future extensions to larger-scale scenarios, such as integrating our approach into an online learning framework to continuously update reliability over time. These extensions represent promising directions for future research.

---

### Official Review · Reviewer_i2fN · 2024-11-01

**Soundness:** 3
**Presentation:** 2
**Contribution:** 3
**Rating:** 5
**Confidence:** 4

**Summary:**

This paper proposes Reliability-Aware RAG, which enhances traditional RAG models by incorporating source reliability into both retrieval and aggregation, reducing the risk of misinformation propagation.

RA-RAG iteratively estimates source reliability and performs inference based on this reliability, achieving improved performance on the proposed benchmark.

**Strengths:**

1. The idea of iteratively reliability estimation is interesting, which provides an effective estimation for source reliability in certain scenarios.
2. The proposed benchmark can be applied to future related studies.

**Weaknesses:**

1. Questioning the effectiveness of the method.

a) The method proposed by the author relies heavily on the correctness of majority voting, both in reliability assessment and during the inference phase. However, in noisy scenarios (for example, when there is only one correct document), this method may not precisely select reliable document. This is precisely an important scenario that the paper aims to address.

b) In my opinion, the core assumption of this method is that the higher the probability of different documents in a certain source providing the same answer, the greater the overall reliability of that source. Furthermore, the author's approach to constructing the benchmark seems to follow this assumption, which could be the reason that the reliability estimation is effectively applicable on the benchmark created in this paper. However. due to the imperfect retrieval and the spread of misinformation, this assumption may not hold true in real RAG systems. I believe the author should validate the effectiveness of this method in more realistic scenarios.

2. Inappropriate experimental settings

a) In Figure 2 and 4, I cannot observe a significant performance improvement compared to the baseline WMV. I strongly recommend that the authors present the experimental results in a table. And the author should provide a comparison with more other relevant methods (such as CAG / RobustRAG).

3. Limitations of the proposed benchmark

• The paper claims to present a realistic multi-source RAG benchmark. However, all the misinformation is generated by GPT-4 mini rather than real-world data. This will limit the demonstration of the method's effectiveness in actual scenarios.

**Questions:**

see Weaknesses

---

> ### Author Response · Authors · 2024-11-17
> **Author Response to Official Review by Reviewer i2fN (1/3)**
>
> #### We sincerely thank you for your detailed and constructive feedback. We truly value the time and effort you’ve invested in reviewing our work. In the following, we address each of your comments.
>
> ## Weakness
>
> > Weakness 1-a. The method proposed by the author relies heavily on the correctness of majority voting, both in reliability assessment and during the inference phase. However, in noisy scenarios (for example, when there is only one correct document), this method may not precisely select a reliable document. This is precisely an important scenario that the paper aims to address.
>
> #### We first want to correct the reviewer’s potential misunderstanding of our method. Our method for reliability estimation and inference phase is based on the weighted majority voting rather than the majority voting.
>
> #### Reliability estimation algorithm iteratively adjusts source weights until convergence, assigning greater weight to more reliable sources and lower weight to less reliable ones.
> In the inference phase, we then use these calculated weights for weighted majority voting.
>
> #### As shown in Figure 3, our RA-RAG method correctly identifies the reliable document even in noisy scenarios where only one document is accurate. Additionally, Figure 4 demonstrates the robustness of RA-RAG compared to majority voting in noisy scenarios as the number of spammers (malicious sources) increases.
>
> > Weakness 1-b. In my opinion, the core assumption of this method is that the higher the probability of different documents in a certain source providing the same answer, the greater the overall reliability of that source. Furthermore, the author's approach to constructing the benchmark seems to follow this assumption, which could be the reason that the reliability estimation is effectively applicable on the benchmark created in this paper. However, due to the imperfect retrieval and the spread of misinformation, this assumption may not hold true in real RAG systems. I believe the author should validate the effectiveness of this method in more realistic scenarios.
> #### In our understanding, your concern lies in that our method might appear to advantageously estimate reliability in cases where a large number of documents from a source provide the same answer, potentially leading to an increase in the overall reliability of that source.
>
> #### However, our reliability estimation is based on questions, ensuring that the number of related documents for each question does not impact the reliability assessment. In RAG systems, the LLM synthesizes multiple retrieved documents to generate a single response, so even if multiple documents provide the same answer, it does not influence the reliability estimation.
>
> #### Additionally, when a source contains documents with conflicting answers, it indicates the source has low reliability. In such cases, the LLM may struggle to produce a correct response due to the mix of accurate and inaccurate information, leading to a lower estimated reliability.
>
> #### To address imperfect retrieval, we employ “I don’t know (IDK)” responses and filtering methods. Specifically,  if no relevant information is available for a given question, we first prompt the LLM to respond with “IDK” (Section 3.1). However, this alone is insufficient due to potential misaligned LLM responses. To mitigate this, we apply a filtering method that replaces the LLM’s response with “IDK” when the response is not grounded in the retrieved documents (Section 3.2). Queries where the filtered LLM response is “IDK” are excluded from both the reliability estimation process and the inference phase. As demonstrated in Table 1, this approach effectively reduces the impact of irrelevant information and misaligned responses.

---

> > ### Comment · Reviewer_i2fN · 2024-11-26
> > **Response for Rebuttal**
> >
> > Dear authors,
> >
> > Thank you for your response! After careful consideration, I have decided to retain the current score.

---

> ### Author Response · Authors · 2024-11-17
> **Author Response to Official Review by Reviewer i2fN (2/3)**
>
> > Weakness 2-a. Inappropriate experimental settings. a) In Figure 2 and 4, I cannot observe a significant performance improvement compared to the baseline WMV. I strongly recommend that the authors present the experimental results in a table. .
>
> #### To clarify, WMV is an ablation of RA-RAG, as it excludes the Reliable and Relevant Source Selection (RRSS) process from RA-RAG. While WMV aggregates outputs from all sources, RA-RAG improves computational efficiency by selecting a subset of sources using $\kappa$-RRSS. Despite using only a small subset of sources, RA-RAG achieves comparable performance to WMV, as shown in Figures 2 and 4.
> #### The experimental results for Figure 2, conducted using the NQ dataset and Llama3-8b, are as follows:
> |                | 3       | 4       | 5       | 6       | 7       | 8       | 9       |
> |----------------|---------|---------|---------|---------|---------|---------|---------|
> | Oracle WMV     | 0.55036 | 0.59321 | 0.6215  | 0.63814 | 0.64764 | 0.70143 | 0.73443 |
> | WMV            | 0.54643 | 0.58957 | 0.61607 | 0.63114 | 0.64493 | 0.70114 | 0.73436 |
> | RA-RAG (Ours)  | 0.54643 | 0.58957 | 0.615   | 0.62507 | 0.63671 | 0.688   | 0.71986 |
> | MV             | 0.50136 | 0.52271 | 0.57586 | 0.59843 | 0.60243 | 0.67786 | 0.7125  |
> | Naive RAG      | 0.47543 | 0.46136 | 0.47943 | 0.48021 | 0.45986 | 0.5055  | 0.52193 |
> | Naive LLM      | 0.27214 | 0.27214 | 0.27214 | 0.27214 | 0.27214 | 0.27214 | 0.27214 |
>
>
>
>
> #### To avoid any confusion, we will explicitly clarify this in the revised manuscript. Additionally, we will present other experimental results in a table format.
>
> > Weakness 2-b. And the author should provide a comparison with more other relevant methods (such as CAG / RobustRAG)
>
> #### As discussed in related work (Appendix A), relevant methods (such as CAG / Robust RAG) have inherent limitations for our benchmark, as detailed below:
>
> - #### CAG determines document reliability based on source authority, categorizing it as either high or low. Since CAG relies on prior knowledge of source reliability, it cannot be applied to our benchmark, where source reliability is unknown. However, source authority, particularly on social media, is often unclear and prone to manipulation. Moreover, its simplistic categorization (low or high) treats reliability scores like 0.6 and 0.9 as equally high, which limits its ability to robustly aggregate answers.
>
> - #### Robust RAG proposes two methods to enhance robustness: Secure Keyword Aggregation and Secure Decoding Aggregation. Secure keyword aggregation relies on the strong assumption that the portion of misinformation in the retrieved documents is minor. This approach depends on the principle of majority voting, overlooking the heterogeneity in source reliability.  Secure Decoding Aggregation leverages token prediction probabilities to evaluate the confidence of an LLM’s response. However, since the truthfulness of documents cannot be reliably determined by an LLM’s internal knowledge, especially for up-to-date information, this approach has significant limitations.
>
> #### To explicitly demonstrate the vulnerability of Robust RAG, we conducted an experiment using the NQ dataset with a spammer-hammer prior. In this setup, we have five sources: three spammers and two hammers. We used Llama-2-7b-chat-hf for our framework.
>
> The results are as follows:
>
> | Method         | EM     |
> |----------------|--------|
> | RA-RAG (ours)  | 0.529  |
> | Robust RAG (Secure Keyword Aggregation)     | 0.328 |
> | Robust RAG (Secure Decoding Aggregation)     | 0.244 |
> #### We appreciate your feedback. We will revise the paper to include a more detailed discussion of related work and incorporate these experimental results to provide additional clarification and further strengthen our contributions.

---

> ### Author Response · Authors · 2024-11-17
> **Author Response to Official Review by Reviewer i2fN (3/3)**
>
> > Weakness 3. Limitations of the proposed benchmark. The paper claims to present a realistic multi-source RAG benchmark. However, all the misinformation is generated by GPT-4 mini rather than real-world data. This will limit the demonstration of the method's effectiveness in actual scenarios.
>
> #### Our benchmark reflects the complexities of real-world scenarios, where sources have heterogeneous reliability, e.g., web data have highly variable quality. It addresses two key challenges:
> - #### By leveraging a controllable prior (beta prior and spammer-hammer prior), our benchmark allows us to simulate diverse levels of source reliability and query relevance, enabling us to replicate the complexities of real-world scenarios.
> - #### Fact-checking is an inherently labor-intensive task, making it challenging to obtain ground-truth for real-world data. This, in turn, complicates the evaluation process.
> #### We will further clarify these points in the revised manuscript. Additionally, to strengthen our method, we plan to explore integrating real-world data in future extensions to better validate our framework.

---

### Official Review · Reviewer_Lq2Y · 2024-11-10

**Soundness:** 2
**Presentation:** 4
**Contribution:** 3
**Rating:** 3
**Confidence:** 3

**Summary:**

These authors aim to address the challenge of heterogeneous source reliability in RAG, and propose a novel framework, Reliability-Aware RAG (RA-RAG), which incorporates source reliability into both the retrieval and aggregation processes.

**Strengths:**

The paper is well-written.
Considering multi-source reliability is innovative and addresses a critical yet often overlooked aspect of RAG methods.

**Weaknesses:**

1. Despite using prompts for constraints, voting remains influenced by expression diversity and synonyms, which creates a frustrating lack of control in practical applications.
2. Generalization issues, such as in multi-hop scenarios and open-ended questions.
3. I question the motivation behind using RAG responses to certain queries about a resource as the basis for assessing that resource's reliability. For instance, a small amount of data does not necessarily indicate unreliability, and it is challenging to distinguish a small amount of correct information when faced with a large volume of outdated or incorrect data. Similarly, as the retriever is a critical component in RAG, the reliability of the retriever itself should not be overlooked when evaluating the reliability of a resource through retrieval-based methods(The authors' method for predicting reliability can still be viewed as a variation of relevance. ).
4. Lack of baseline to experimentally demonstrate the effectiveness.

**Questions:**

In practice, would this reliability prediction result in multiplied API consumption and reduced efficiency?
It is necessary to compare with some advanced RAG methods to validate the necessity of "considering multi-source reliability".

---

> ### Author Response · Authors · 2024-11-17
> **Author Response to Official Review by Reviewer Lq2Y (1/2)**
>
> #### Thank you for taking the time to review and engage with our paper. We have carefully considered your feedback and questions. Below, we have provided detailed responses to address each of your comments.
> ## Weakness
> > Weakness 1. Despite using prompts for constraints, voting remains influenced by expression diversity and synonyms, which creates a frustrating lack of control in practical applications
> ####  Aggregating responses from LLMs is a fundamental challenge across multiple domains, including reasoning tasks [4, 5] and uncertainty estimation in LLM outputs [1, 2, 3, 6].
> #### To illustrate the core concept of our framework, we simply employ a keyword-based system prompt as a proof of concept in this work.  As a more robust approach, we can incorporate bidirectional entailment clustering into our framework, as suggested in [3, 6].
> #### This method employs the natural language classifier or LLM to perform a bidirectional comparison of responses, determining mutual entailment, i.e., whether response A entails response B and vice versa. If mutual entailment is established, the responses are considered semantically equivalent, enabling more accurate aggregation of diverse expressions based on their semantic content.
> #### We will revise the paper to provide further details and additional experimental results before the end of the rebuttal period.
> > Weakness 2. Generalization issues, such as in multi-hop scenarios and open-ended questions.
> #### Thank you for the interesting suggestions for future work. Since short-form Q&A serves as a foundational task, extending it to multi-hop scenarios and open-ended questions is feasible. We are currently working on extending our method to handle open-ended questions, where responses are in long-form.
> #### Specifically, a long-form response can be decomposed into a set of factoids [5], with each factoid representing a short-form sentence. For each factoid, we can generate questions to which that factoid might have been the answer. Then, weighted majority voting is applied to obtain reliable outputs for each question. Finally, the LLM combines these questions and their corresponding answers to generate the long-form response.
>
> #### Similarly, we believe similar extensions, such as multi-hop Q&A, are feasible and represent promising directions for future research.
>
>  #### We will revise the paper to provide more detailed explanations of this extension and include additional experimental results to validate its effectiveness before the end of the rebuttal period.
>
>
> > Weakness 3-a. I question the motivation behind using RAG responses to certain queries about a resource as the basis for assessing that resource's reliability. For instance, a small amount of data does not necessarily indicate unreliability, and it is challenging to distinguish a small amount of correct information when faced with a large volume of outdated or incorrect data.
>
> #### In our understanding, your concern lies in the potential misestimation of source reliability, when the query for estimating reliability can not be representative of the source.
>
> #### We address this in two ways:
> - #### A source can be defined with our objectives. For instance, instead of treating an entire platform or website as a source, we can define sources at a more granular level, such as individual user accounts or specific communities. This approach allows us to estimate the reliability of sources using certain queries that align with the relevant content of these sources.
> - #### Our approach can be extended to an online learning framework that updates source reliability over time, without explicit labels, which would help address changes in source quality and increase robustness. Those extensions are conceptually possible but pose challenging engineering problems, such as computational efficiency, which could be addressed in future work.
>
>
> > Weakness 3-b. Similarly, as the retriever is a critical component in RAG, the reliability of the retriever itself should not be overlooked when evaluating the reliability of a resource through retrieval-based methods.
>
> #### As we also mentioned in Section 3.1 and 3.2, irrelevant retrieved documents can hinder the LLM and negatively impact reliability estimation. In fact, our methods have addressed these issues.
>
> #### Specifically,  if no relevant information is available for a given question, we first prompt the LLM to respond with “I don’t know (IDK)” (Section 3.1). However, this alone is insufficient due to potential misaligned LLM responses. To mitigate this, we apply a filtering method that replaces the LLM’s response with “IDK” when the response is not grounded in the retrieved documents (Section 3.2). Queries where the filtered LLM response is ``IDK” are excluded from the reliability estimation process.
>
> #### As demonstrated in Table 1, this approach effectively reduces the impact of irrelevant information and misaligned responses, improving the accuracy of reliability estimation.

---

> > ### Comment · Reviewer_Lq2Y · 2024-11-20
> >
> > Thank you for addressing the concerns I raised. First, you mentioned your approach to handling irrelevant documents retrieved during the voting process. However, in the voting stage, relevant documents play a decisive role. I assume your hypothesis is that the retriever can at least retrieve enough relevant information to ensure a fair and information-equivalent voting process (please correct me if my understanding is inaccurate). To support this assumption, experimental evidence such as the retriever’s F1 score for golden references would be helpful.
> >
> > Second, regarding your additional solution to address Weakness 3-a, while I understand that time constraints during the rebuttal phase make additional experiments challenging, the proposed solution does not convincingly demonstrate feasibility or effectiveness. I still consider Weakness 3-a a significant concern.
> >
> > Alternatively, I would be willing to reconsider my score if you could provide further clarification or experimental results on the following:
> >
> > The cost analysis in the paper is overly simplistic and does not adequately demonstrate the method's efficiency or practical applicability. This raises concerns that the proposed additional solution might further increase complexity and cost.
> > Can you provide a statistical comparison of the proposed method against the best baseline in terms of API call counts, token throughput, and runtime? This would help alleviate concerns about scalability and economic feasibility.

---

> ### Author Response · Authors · 2024-11-17
> **Author Response to Official Review by Reviewer Lq2Y (2/2)**
>
> > Weakness 4. Lack of baseline to experimentally demonstrate the effectiveness.
> > Question. It is necessary to compare with some advanced RAG methods to validate the necessity of "considering multi-source reliability".
>
> #### As discussed in related work (Appendix A), even advanced RAG systems such as Self-RAG [7] and CRAG [8] are highly vulnerable to misinformation. This is because existing RAG systems rely on similarity scores for document retrieval, so they have a risk of incorporating relevant but unreliable content only from malicious sources, as demonstrated in PoisonedRAG [9].
>
> #### To explicitly demonstrate this vulnerability, we conducted an experiment using the NQ dataset with a spammer-hammer prior. In this setup, we have five sources: two spammers and three hammers. We used Llama-2-7b-chat-hf for our framework. The results are as follows:
>
> | Method         | EM     |
> |----------------|--------|
> | RA-RAG (ours)  | 0.598  |
> | self-RAG       | 0.388  |
>
> #### These results demonstrate the critical importance of incorporating multi-source reliability into retrieval-augmented generation frameworks.
> #### We appreciate your feedback and will revise the paper to include these experimental results to provide further clarification and strengthen our contributions.
> ## Question
>
> > Question 1. In practice, would this reliability prediction result in multiplied API consumption and reduced efficiency?
>
> #### While incorporating multi-source reliability estimation may introduce some additional computational cost, we consider it a necessary trade-off for achieving more robust and reliable outcomes. As demonstrated by our experimental results above, RA-RAG significantly outperforms Self-RAG in scenarios with malicious sources, highlighting the importance of the multi-source RAG framework.
>
>
> #### [1] Ott, Myle, et al. "Analyzing uncertainty in neural machine translation." ICML 2018.
> #### [2] Sai, Ananya B., Akash Kumar Mohankumar, and Mitesh M. Khapra. "A survey of evaluation metrics used for NLG systems." ACM Computing Surveys (CSUR) 55.2 (2022): 1-39.
> #### [3] Kuhn, Lorenz, Yarin Gal, and Sebastian Farquhar. "Semantic Uncertainty: Linguistic Invariances for Uncertainty Estimation in Natural Language Generation." ICRL 2023.
> #### [4] Wang, Xuezhi, et al. "Self-Consistency Improves Chain of Thought Reasoning in Language Models."   ICRL 2023.
> #### [5] Yao, Shunyu, et al. "ReAct: Synergizing Reasoning and Acting in Language Models."  ICRL 2023.
> #### [6] Farquhar, S., Kossen, J., Kuhn, L. et al. Detecting hallucinations in large language models using semantic entropy. Nature 630, 625–630 (2024).
> #### [7] Asai, Akari, et al. "Self-RAG: Learning to Retrieve, Generate, and Critique through Self-Reflection."  ICRL 2024.
> #### [8] Yan, Shi-Qi, et al. "Corrective retrieval augmented generation." arXiv preprint arXiv:2401.15884 (2024).
> #### [9] Zou, Wei, et al. "Poisonedrag: Knowledge corruption attacks to retrieval-augmented generation of large language models." USENIX Security (2025).

---

> ### Author Response · Authors · 2024-11-23
> **Author Response to Official Review by Reviewer Lq2Y**
>
> #### We express our deep appreciation for your time and insightful comments. In what follows, we address each of your comments.
>
> > Question 1. However, in the voting stage, relevant documents play a decisive role. I assume your hypothesis is that the retriever can at least retrieve enough relevant information to ensure a fair and information-equivalent voting process (please correct me if my understanding is inaccurate). To support this assumption, experimental evidence such as the retriever’s F1 score for golden references would be helpful.
>
>
> #### Under our setting of $r_i=0.6$ (the probability of sources containing relevant documents for a given query), the retriever achieves an R@1 0.90, R@2 0.94, R@3 0.96 for valid queries for which a source has relevant documents. (R@K indicates that there is at least one relevant document in the top-K retrieved documents). These results show that the retriever performs well in retrieving relevant documents. Additionally, for all queries, the retriever achieves an R@1 0.55, R@2 0.56, R@3 0.58. This indicates that we can control the source relevance using $r_i$.
>
> #### We want to clarify that our reliability estimation algorithm does not require all sources to participate in voting. For stability, we exclude the cases where the voting is not established, e.g., only one or two sources contain relevant documents, or where ties occur.
> > Question 2. Second, regarding your additional solution to address Weakness 3-a, while I understand that time constraints during the rebuttal phase make additional experiments challenging, the proposed solution does not convincingly demonstrate feasibility or effectiveness. I still consider Weakness 3-a a significant concern.
>
> #### The size of the query set for reliability estimation depends on the size of the source and the relevance between sources. In this study, the source size is relatively small (granular source), consisting of approximately 2,800 documents with $r_i=0.6$ and a predefined set of queries for reliability estimation is available. This allows us to use a small set of queries for reliability estimation. However, as the source size increases and the relevance between sources decreases, a larger number of queries would be required for accurate reliability estimation.
> #### We would like to emphasize that this work serves as a foundational study demonstrating the feasibility of incorporating source reliability estimation into RAG systems to enhance their robustness against misinformation. By addressing a small-scale scenario, our framework lays the groundwork for future extensions to larger-scale scenarios, such as integrating our approach into an online learning framework to continuously update reliability over time. These extensions represent promising directions for future research.
>
> > Question 3. Alternatively, I would be willing to reconsider my score if you could provide further clarification or experimental results on the following:
> The cost analysis in the paper is overly simplistic and does not adequately demonstrate the method's efficiency or practical applicability. This raises concerns that the proposed additional solution might further increase complexity and cost. Can you provide a statistical comparison of the proposed method against the best baseline in terms of API call counts, token throughput, and runtime? This would help alleviate concerns about scalability and economic feasibility.
> #### To demonstrate the efficiency of our method, we conducted an experiment on the NQ dataset using a beta prior with nine sources, as described in Section 5.2. For this experiment, we utilized the GPT-4o-mini API. The experimental results are as follows:
> | Method | API Call (Reduction %) | Token Throughput (Improvement %) | Run Time (Reduction %) |
> |--------|-------------------------|---------------------------|-------------------------|
> | RA-RAG (ours)   | 7080 (-43.81%)          | 997 (+14.34%)             | 4418 (-50.91%)          |
> | WMV    | 12600 (0.00%)          | 872 (0.00%)               | 8999 (0.00%)           |
> #### The experimental results demonstrate that our method reduces API calls, and runtime and improves token throughput compared to weighted majority voting (WMV). While WMV incurs high computational costs using all sources, RA-RAG uses the k-RRSS process to select a smaller subset of sources, reducing computational overhead.

---

> > ### Comment · Reviewer_Lq2Y · 2024-11-24
> >
> > Thanks for your response. I am a bit confused about the units of the numbers in this table. For example, the units for API calls could be "times per task," the throughput could be "number of tokens per task," and the runtime could be "seconds per task."

---

> ### Author Response · Authors · 2024-11-24
> **Author Response to Official Review by Reviewer Lq2Y**
>
> #### We sincerely apologize for the confusion and deeply appreciate your time and thoughtful consideration. The units for the metrics in the table are as follows:
>
> - #### API Call: The total number of API requests made across all tasks. Specifically, this refers to the total number of API calls required for the experiment.
> - #### Token Throughput: The total number of tokens processed (input tokens + output tokens) divided by the Runtime.
> - #### Runtime: The total execution time in seconds for all tasks. Specifically, this represents the total execution time (seconds) for the experiment.

---

> > ### Comment · Reviewer_Lq2Y · 2024-11-25
> >
> > Why is throughput not measured in terms of a single API call or a single QA task? Similarly, runtime and API calls are not measured on a per-task basis but rather on an entire experiment basis. I'm sorry, but I can't gain a deeper understanding of GIVE's efficiency and cost aspects from this result.

---

> ### Author Response · Authors · 2024-11-26
> **Author Response to Official Review by Reviewer Lq2Y**
>
> #### We sincerely apologize for the confusion. To address the confusion, we have conducted additional experiments on a per-task basis. Below, we clarify the definitions of the metrics in the updated table. The experimental results are averaged for all tasks:
>
> - #### API Call: The total number of API requests for a single task.
> - #### Token Throughput:  The total number of tokens processed (input tokens + output tokens) divided by the runtime for a single task.
> - #### Runtime: The total execution time, measured in seconds, required for a single task.
>
>
> | Method | API Call (Reduction %) | Token Throughput (Improvement %)| Run Time (Reduction %) |
> |--------|-------------------------|---------------------------|-------------------------|
> | RA-RAG (ours)   | 6.495 (-27.83 %)        | 993.4 (-0.06%)      | 4.391  (-25.55 %)        |
> | WMV    | 9.0         | 999.4            | 5.898           |
>
> #### RA-RAG demonstrates slightly better computational efficiency than WMV in this setting. While the efficiency gains from the $\kappa$-RRSS process are modest with a small number of sources, the advantages become more pronounced as the number of sources increases. To demonstrate this, we have conducted additional experiments with 20 sources under the same experimental setup. The results are as follows:
>
> | Method | API Call (Reduction %) | Token Throughput (Improvement %) | Run Time (Reduction %) |
> |--------|-------------------------|---------------------------|-------------------------|
> | RA-RAG (ours)   |  12.52 (-37.4 %)     | 948.17 (+0.21 %)          | 12.98 (-34.97 %)          |
> | WMV    | 20          | 946.16              | 19.96           |
>
> #### These results demonstrate that RA-RAG achieves greater computational efficiency than WMV as the number of sources increases. Furthermore, as the number of sources continues to grow, RA-RAG is expected to achieve larger efficiency gains.

---

> ### Comment · Reviewer_Lq2Y · 2024-11-26
>
> 1) A new concern about the validity of these results: the numbers of tasks (Total API Calls / Avg. API Calls) tested for RA-RAG and WMV are inconsistent.  All runtime results are also inconsistent. Please provide detailed experimental settings (e.g. the number of tasks tested) and explain.
>
> 2) Still concern Weakness 3-a given that the experimental results failed to clearly present the efficiency and cost issue of the proposed method.
>
> Thank you for your answers.

---

> ### Author Response · Authors · 2024-11-26
> **Author Response to Official Review by Reviewer Lq2Y**
>
> #### In our understanding, a per-task basis measures the cost required to solve a single task (a question in Q&A dataset) for each method. For example, the API Call represents the number of API calls needed to solve a single question, and Runtime measures the time taken to solve a single question.
>
> #### For both RA-RAG and WMV, we evaluated API Call, Token Throughput, and Runtime using the same test dataset. Since these metrics can vary across tasks (e.g., some questions may require more API calls or runtime), we report the average values across all tasks in the dataset.
>
> #### Specifically, in the first experimental table, WMV requires 9 API calls as it retrieves documents from all 9 sources, whereas RA-RAG requires an average of 6.495 API calls due to its dynamic source selection. Additionally, WMV takes 5.898 seconds to solve a task, while RA-RAG takes 4.391 seconds.
>
> #### We have observed that runtime is not consistent; running the same code for the same question can result in significantly different execution times due to external factors.
>
> #### If my understanding of the metrics is inaccurate, could you kindly provide more detailed clarification?

---

### Author Response · Authors · 2024-11-23
**Author Response to All Reviewers**

#### Dear Reviewers,

#### Thank you for your thorough engagement and constructive feedback on our paper. We would like to follow up to confirm that we have made the changes for clarification and incorporated them into the paper. In our updated manuscript, we highlight the changes in pink. Additionally, we have included additional experiments in the Appendix:

#### - Advanced aggregation method: Following the comments of Reviewers Lq2Y and AMyi, we have implemented an advanced aggregation method to improve robustness in handling linguistic variation. Details can be found in Appendix I.
#### - Extension for long-form generation: Following the suggestion from Reviewer Lq2Y, we have extended our framework for long-form generation. This extension is detailed in Appendix J.

---

### Meta-Review · Area_Chair_VNxL · 2024-12-24

**Metareview:**

This paper introudces a new RAG framework, named reliability-aware RAG (RA-RAG), which estimates the reliability of multiple sources and incorporates this information into both the retrieval and aggregation processes. Moreover, the authors also introduce a benchmark dataset designed to reflect the real scenarios with heterougeneous source reliability and demonstrate the effectiveness of the proposed method.

Overall, this paper studies an interesting problem. However, the motivation of the proposed method is not well explained. The experimental analysis is not sufficient, lacking relevant baselines to demonstrate the effectiveness of the proposed method. Moreover, the application of the proposed method in real scenarios is limited.

**Additional Comments On Reviewer Discussion:**

In the rebuttal, the authors provide more experimental results of baseline methdos, e.g., self-RAG, WMV, Robust RAG. They also include more analysis about the cost of the proposed method, in terms of API call counts, token throughout, and runtime. The authors have addressed some concerns of the reviewers. However, the reviewers' concerns regarding with the motivation of using RAG to assess resource reliability through query responses and the inconsistency of some experimental results still remain.

---

### Decision · Program_Chairs · 2025-01-22

Reject